

# Snow Microphysical Retrieval from the NASA D3R Radar During ICE-POP 2018

S. Joseph Munchak[1,*], Robert S. Schrom[1,2], Charles N. Helms[1,2], and Ali Tokay[1,3]

[1]Mesoscale Atmospheric Processes Laboratory, NASA Goddard Space Flight Center, Greenbelt, MD, USA
[*]now at The Tomorrow Companies, Inc.
[2]Universities Space Research Association, Columbia, MD, USA
[3]Joint Center for Earth Systems Technology, University of Maryland, Baltimore County, Catonsville, MD, USA

**Correspondence:** Robert S. Schrom (robert.s.schrom@nasa.gov)

**Abstract.**

A method is developed to use both polarimetric and dual-frequency radar measurements to retrieve microphysical properties of falling snow. It is applied to the Ku- and Ka-band measurements of the NASA Dual-polarization, Dual-frequency Doppler Radar (D3R) obtained during the International Collaborative Experiment for PyeongChang Olympic and Paralympics

(ICE-POP 2018) field campaign, and incorporates the Atmospheric Radiative Transfer Simulator (ARTS) microwave single scattering property database for oriented particles. The retrieval uses optimal estimation to solve for several parameters that describe the particle size distribution (PSD), relative contribution of pristine, aggregate, and rimed ice species, and the orientation distribution along an entire radial simultaneously. Examination of Jacobian matrices and averaging kernels show that the dual wavelength ratio (DWR) measurements provide information regarding the characteristic particle size, and to a lesser extent,

the rime fraction and shape parameter of the size distribution, whereas the polarimetric measurements provide information regarding the mass fraction of pristine particles and their characteristic size and orientation distribution. Thus, by combining the dual-frequency and polarimetric measurements, some ambiguities can be resolved that should allow a better determination of the PSD and bulk microphysical properties (*e.g.*, snowfall rate) than can be retrieved from single-frequency polarimetric measurements or dual-frequency, single-polarization measurements.

The D3R ICE-POP retrievals were validated using Precipitation Imaging Package (PIP) and Pluvio weighing gauge measurements taken nearby at the May Hills ground site. The PIP measures the snow PSD directly, and its measurements can be used to derived the snowfall rate (volumetric and water equivalent), mean volume-weighted particle size, and effective density, as well as particle aspect ratio and orientation. Four retrieval experiments were performed to evaluate the utility of different measurement combinations: Ku-only, DWR-only, Ku-pol, and All-obs. In terms of correlation, the volumetric snowfall rate

($r = 0.95$) and snow water equivalent rate ($r = 0.92$) were best retrieved by the Ku-pol method, while the DWR-only method had the lowest magnitude bias for these parameters (-31% and -8%, respectively). The methods that incorporated DWR also had the best correlation to particle size ($r = 0.74$ and $r = 0.71$ for DWR-only and All-obs, respectively), although none of the methods retrieved density particularly well ($r = 0.43$ for All-obs). The ability of the measurements to retrieve mean aspect ratio was also inconclusive, although the polarimetric methods (Ku-pol and All-obs) had reduced biases and MAE relative to the





Ku-only and DWR-only methods. The significant biases in particle size and snowfall rate appeared to be related to biases in the measured DWR, emphasizing the need for accurate DWR measurements and frequent calibration in future D3R deployments.

## 1 Introduction

Estimation of snowfall rates and other properties from weather radar is made difficult by many of the same challenges that exist for rainfall estimation (primarily, the discrepancy between the 6th moment dependence of radar reflectivity factor $Z$
and the 3rd-4th moment dependence of precipitation rate $R$), but additional factors further confound radar retrievals of snow. Whereas the shape and scattering properties of a raindrop depend only on its mass and temperature (e.g., Beard et al., 2010; Ekelund et al., 2020), there is tremendous diversity in ice crystals of a given mass, resulting from the infinite complexity of particle trajectories through differing thermodynamic environments resulting in growth by vapor deposition (e.g., Kuroda and Lacmann, 1982; Chen and Lamb, 1994; Fukuta and Takahashi, 1999), aggregation (e.g., Hosler et al., 1957; Hobbs et al.,
1974; Connolly et al., 2012), riming (e.g., Mitchell et al., 1990; Jensen and Harrington, 2015; Moisseev et al., 2017), as well as ablation by sublimation (e.g., Smith et al., 2009) and melting (e.g., Matsuo and Sasyo, 1981; Leinonen and von Lerber, 2018). All of these processes influence the scattering and aerodynamic properties of these ice particles in ways that can influence the interpretation of radar data (e.g., Hall et al., 1984; Vivekanandan et al., 1994; Bechini et al., 2013; Botta et al., 2013; Thompson et al., 2014). Despite these challenges, since the introduction of weather radar, these instruments have been important tools
in gathering information about ice-phase precipitation. Early efforts focused on using $Z$ to estimate the intensity of snow precipitation measurements with assumed ice particle size distribution (PSD) forms (e.g., Marshall and Gunn, 1952). These efforts relied on *in situ* ground measurements to derive empirical relations to the measured Z, yielding a variety of Z-S relations, depending on the climatological properties of snow at the measurement location.

Improvements upon these situational Z-S relationships can be made if multiparameter radar observations are available.
For snow, these have historically proceeded along two pathways following advances in multi-frequency/Doppler and dual-polarization radar technologies. Multi-frequency methods essentially rely upon deviations from Rayleigh scattering to infer a characteristic particle size (e.g., Matrosov et al., 2005; Liao et al., 2016) and, with three frequencies (typically X or Ku, Ka, and W bands), density can also be inferred (Kneifel et al., 2015). For vertically-pointing radars, Doppler velocities can also be used to refine the density estimate (Oue et al., 2015; Mason et al., 2018), since fall velocity of a snow particle depends (to first
order) on its size and density (Heymsfield and Westbrook, 2010). These methods have been employed primarily towards data collected at a few well-equipped snow observatories such as the Hyytiälä Forestry Research Station in Hyytiälä, Finland (Hari and Kulmala, 2005), the Jülich Observatory for Cloud Evolution (JOYCE) in Germany (Löhnert et al., 01 Jul. 2015), and the Department of Energy - Atmospheric Radiation Measurement Program facility in Alaska (de Boer et al., 2018) and applied to airborne and spaceborne radar datasets (Leinonen et al., 2018; Tridon et al., 2019; Chase et al., 2021).
Polarimetric radar measurements have shown value in inferring ongoing ice growth processes, owing to the dependence of these measurements on the distributions of particle shapes, orientations, and sizes. In particular, enhancements in the differential reflectivity ($Z_{DR}$) and specific differential phase ($K_{dp}$) have been linked to the planar crystal growth near -15 °C (e.g., Ryzhkov





and Zrnić, 1998; Kennedy and Rutledge, 2011; Andrić et al., 2013; Schrom et al., 2015; Moisseev et al., 2015). Assessing changes in vertical profiles of the polarimetric radar variables also provides information on ice growth processes. Decreases

in $Z_{DR}$ and $K_{dp}$ towards the ground have been observed with increases in reflectivity towards the ground, indicating growing ice particles becoming more chaotically oriented and more spherical, a result of some combination of aggregation and intense riming (e.g., Bechini et al., 2013; Oue et al., 2015; Ryzhkov et al., 2016; Schrom and Kumjian, 2016; Kumjian and Lombardo, 2017). Some recent efforts have been made to gain quantitative information about the ice particle properties (and thus associated microphysical processes and snowfall rates) from polarimetric radar measurements using empirically determined algorithms

(e.g., Bukovčić et al., 2020) and microphysical-model informed parameter estimation (e.g., Schrom et al., 2021). However, there has been limited evaluation of these methods using additional remote sensing (e.g., multi-frequency radar measurements) and *in situ* observations.

From the extensive literature on multi-frequency and polarimetric radar studies of snow, it is evident that complementary information is contained in these measurements. However, relatively few studies have been performed to assess the informa-

tion content of multi-frequency, dual-polarization radar measurements of snow, partially because of a lack of radar platforms with these capabilities deployed in locations subject to frequent and high-accumulation snowfall events. The NASA Dual-polarization, Dual-frequency Doppler Radar (D3R) is a premier radar for making such measurements. The D3R was built to provide ground validation measurements for the NASA Global Precipitation Measurement (GPM) mission's Dual-Frequency Precipitation Radar (DPR; Chandrasekar et al., 2010; Vega et al., 2014), and operates at Ku and Ka bands using novel solid

state transmitters. The D3R was first deployed in the GPM Cold-season Precipitation Experiment (GCPex; Skofronick-Jackson et al., 2015) and in subsequent GPM ground validation field campaigns. Upgrades to improve sensitivity and range resolution have been implemented (Kumar et al., 2017) since these early campaigns.

Recognizing the potential utility of D3R measurements to provide unique information about microphysics, dynamics, and quantitative precipitation estimation (QPE) in snowstorms, the Korean Meteorological Administration (KMA), organizers of

the International Collaborative Experiment for PyeongChang Olympic and Paralympics (ICE-POP 2018), cooperated with NASA to deploy the D3R radar in Daegwallyeong, South Korea from November 2017-March 2018. The D3R was part of an extensive network of ground-based remote sensing and *in-situ* instrumentation deployed during ICE-POP 2018, and formed a central observation point for measurements aligned perpendicular to the coastal mountain ranges of eastern South Korea. This measurement strategy was devised to examine the distribution of precipitation from the coast to the mountains in different

winter synoptic weather situations and evaluate high-resolution numerical weather prediction in this complex topographic region (Lim et al., 2020).

The objective of this study is to use the data collected by the D3R during ICE-POP 2018 develop a snow retrieval algorithm using realistic scattering models of pristine, aggregate, and rimed snow particles, to further our understanding of the complementary nature of the dual-frequency and polarimetric radar measurements and their utility regarding snow microphys-

ical characterization and QPE. The output of this algorithm is intended to aid in identifying microphysical processes during ICE-POP events and provide snow QPE during the deployment. The extensive network of ground instrumentation is leveraged to validate the algorithm output. The manuscript is organized as follows: the observational datasets are listed in section 2,





the particle scattering properties we use and the construction of lookup tables from these databases is described in section 3,
algorithm mechanics and information content are analyzed in section 4, validation for selected cases is presented in section 5,
and the conclusions are given in section 6.

## 2 Datasets

### 2.1 D3R

The NASA D3R radar is a polarimetric Doppler weather radar operating at Ku (13.91 GHz) and Ka (35.56 GHz) bands, which
utilizes novel design features including aligned antennas, solid-state transceivers, and a digital waveform generator to enable
deployment in a wide range of environmental conditions on a mobile trailer platform (Chandrasekar et al., 2010). At both
frequencies, the following parameters are measured: Reflectivity ($Z$), Differential Reflectivity ($Z_{dr}$), Differential Propagation
Phase ($\phi_{dp}$), Co-polar Correlation Coefficient ($\rho_{hv}$), Radial Velocity ($V$), and Spectrum Width ($W$).

During ICE-POP 2018, the D3R was located on the roof of the Daegwallyeong (DGW) Regional Weather Office (36.677°N,
128.719°E, altitude 789m msl). The D3R was configured to measure 150m range gates out to a maximum range of 39.75
km, combining a short pulse for ranges $< 3.3$km with a medium pulse for the remaining range gates. This gives a Ku-band
sensitivity ranging approximately from -30 to -5 dBZ over the short pulse, and from -15 to 5 dBZ over the long pulse (Kumar
et al., 2017). The primary scan schedule during snow events conducted a PPI scan at 5° elevation follow by RHI scans at
51°, 231°, and 330° azimuths, with each set of scans taking 5 minutes. For this study, we focus on the 231° RHI scans aimed
towards the May Hills Supersite (MHS) 2 km downrange, which contained a wealth of ground instrumentation.

We use the the D3R data available from the NASA ICE-POP data archive held at the Global Hydrology Resource Center
(Petersen et al., 2018). Several snowfall events were observed by D3R during the ICE-POP 2018 campaign, but we selected
only three events to analyze in this manuscript, listed in Table 1, for their diversity in synoptic forcing, environmental profiles,
and microphysics. We disregarded events that had mixed-phase precipitation, as the modeling of their scattering properties is
less mature than ice particles for our needs, as well as events that did not have both the D3R and data from ground instruments
at MHS available. The D3R $Z_{dr}$ and $\phi_{dp}$ were calibrated on an event basis, and the absolution calibration of Ku and Ka $Z$ was
established in the early period of the deployment (Chandrasekar et al., 2018). Notwithstanding these calibration efforts, in order
for the retrieval algorithm to perform optimally, we examined the data for self-consistency of $Z_{dr}$ and the Dual-Wavelength
Ratio (DWR), defined as $Z_{Ku} - Z_{Ka}$. The expectation is that, at near-zenith angles, $Z_{dr}$ should be close to zero, and, in the
limit of small particles, the DWR should equal the difference in attenuation, which should be small, but positive, depending on
the path-integrated attenuation from water vapor, cloud liquid water, and hydrometeors.

We examined time series of the pdfs of these quantities for the events listed in Table 1. In Figure 1, the top two rows show
the time series of the pdf of Ku and Ka $Z_{dr}$ at elevation angles $> 80°$ and altitudes above 1.5km msl. There is clearly some
non-stationary behavior in these pdfs, so we applied additional calibration offsets (independently at each frequency) to the
$Z_{dr}$ for each RHI such that the average $Z_{dr}$ at elevation angles $> 80°$ is equal to zero. The time series of the calibrated Ku
and Ka $Z_{dr}$ are shown in the third and fourth rows of Figure 1. In addition to the modification to $Z_{dr}$, we also noticed some





**Table 1.** Selected snowfall events during ICE-POP 2018. Synoptic classification is from Kim et al. (2021).

| Start Date/Time - End Date/Time (UTC) | Synoptic Classification | PIP Snow (cm) | Pluvio SWER (mm) |
|---|---|---|---|
| 09 Jan 2018/1320UTC - 09 Jan 2018/2016UTC | Cold Low | 2.51 | 1.10 |
| 28 Feb 2018/0306UTC - 28 Feb 2018/1620UTC | Warm Low | 63.77 | 58.65 |
| 07 Mar 2018/0949UTC - 08 Mar 2018/1857UTC | Warm Low | 17.38 | 17.32 |

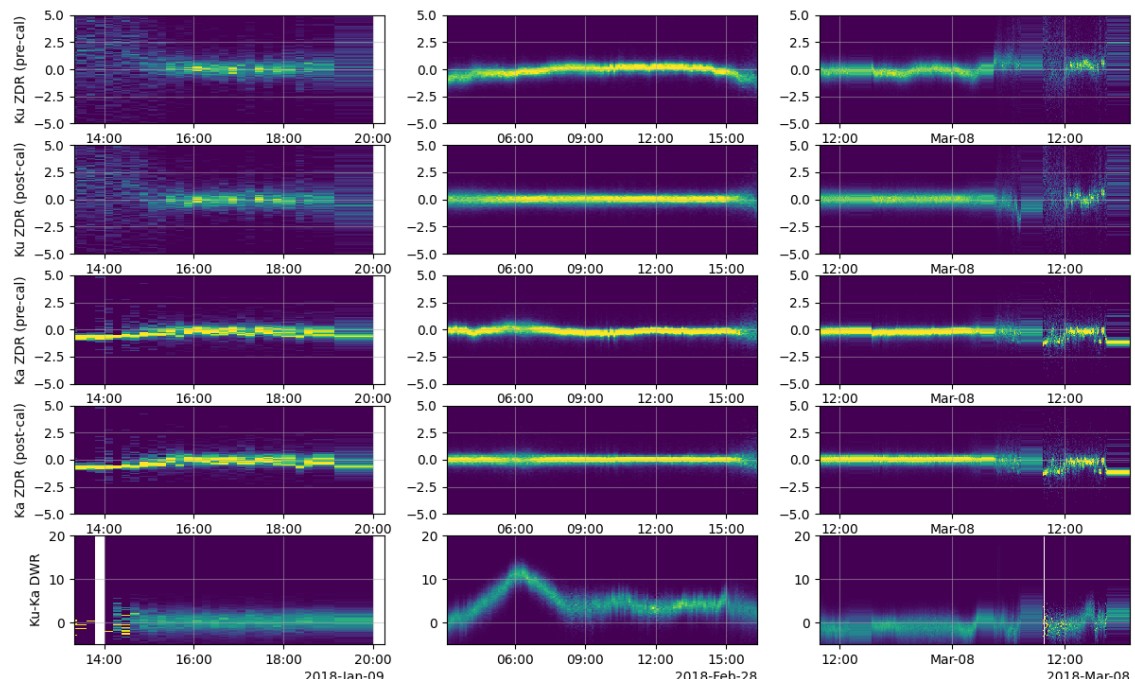

**Figure 1.** Time-series histograms of near-zenith Ku $Z_{dr}$ before (top row) and after (second row) offsets were applied. Similar histograms are presented for Ka-band data in rows three and four. The fifth row displays the time series histogram of the Ku-Ka Dual-Wavelength Ratio where Ku reflectivity is less than 10 dBZ within 10 km of the radar. Colors are indicative of relative occurrence, normalized at each time step.

unusual behavior in the small-particle DWR pdf (fifth row of Figure 1) during the February 28 case. When $Z_{Ku} < 10$dBZ, Skofronick-Jackson et al. (2019) found that Ku-Ka DWR is typically close to zero, but during this case, there is large increase in the DWR, peaking around 0600 UTC, that is difficult to explain entirely by particle size or attenuation. Accumulation of wet snow on the radome is a possible explanation for this behavior (V. Chandrasekahr, pers. comm.), but for this study we have





chosen to avoid correcting the DWR because it is difficult to have a continuous independent estimate of the Ku-Ka relative calibration from a ground radar.

## 2.2 PIP

The primary source of validation for the microphysical retrievals in this study is the Precipitation Imaging Package (PIP; Pettersen et al., 2020). The PIP is a video disdrometer made up of a single high-speed camera, continuously recording at

380 frames per second, and a halogen lamp, which is used to backlight the precipitation particles. The camera and lamp are separated by 2 m and the focal plane is located 1.33 m from the camera lens and uses an open sampling volume (i.e., the sampling volume is not enclosed within a box). The images produced are 640 by 480 pixels with a resolution of 0.1 mm by 0.1 mm. The field of view (FOV), including the edge effects, is then $64 - D_{eq}$ mm by $48 - D_{eq}$ mm, where $D_{eq}$ is the equivalent diameter in millimeters. The depth of field (DOF) also varies with particle size and is expressed as $117/D_{eq}$ (in mm). The

sampling volume is a multiplication of the FOV, DOF, and the number of frames over a given time period. Considering 100 particles with uniform size of $D_{eq} = 1$ mm, the sampling volume is 790 m$^3$ for a one minute observation period. As PIP only uses a single camera, the precipitation particle images are of a projected view of the particle and do not contain any information on the particle dimension along the viewing direction.

PIP determines the characteristics of the precipitation particles using an algorithm written using the National Instruments

IMage AQuisition (IMAQ) software package. This algorithm determines the shape of the precipitation particle (i.e., the long and the short dimensions) by fitting an ellipse to the PIP-imaged particle. The IMAQ software package defines the fitted ellipse as the ellipse having both the same area and the same perimeter as the PIP-imaged particle. During our preliminary analysis of the data, we found that the IMAQ-fitted ellipses tended to overestimate the long dimension of the particle and underestimate the short dimension, resulting in an underestimate of the aspect ratio. As such, we have reprocessed the PIP data using an

alternative, custom-built ellipse fitting strategy. This strategy uses the method implemented in the fit_ellipse program in the Coyote IDL Library (http://www.idlcoyote.com) to perform the actual fit. The fitting is performed on images of particles taken from videos that PIP records for troubleshooting purposes. These videos contain the first 2000 frames that contain precipitation particles within each 10 minute period. For periods with fewer than 2000 frames containing precipitation particles, the videos will be shorter than 2000 frames.

The PIP-determined orientation angle is defined as the counterclockwise angle from the positive horizontal axis, where positive is to the right, to the longest dimension of the particle. This results in orientation angles ranging from 0° to 180°. In order to combine the ellipse aspect ratio (minor axis length divided by major axis length) and orientation angle information, we have used the natural logarithm of the aspect ratio of the bounding box of the particle. The bounding box is defined as the smallest rectangle that is able to contain the particle and whose edges are either horizontally or vertically oriented.

## 2.3 Soundings

Radiosonde launches were performed every three hours during precipitation events from the DGW Regional Weather Office, supplementing the normal 12-hourly observations, and used Meteomodem M10 radiosondes (Meteomodem, cited 2021). The





**Table 2.** Properties of the ARTS scattering database calculations used herein.

| Particle type | $D_{\text{eqv}}$ (mm) | # sizes | $\beta$ (°) | # orientations |
|---|---|---|---|---|
| Pristine plate (ID9) | 0.010–2.596 | 51 | 0–90 | 10 |
| Large plate aggregate (ID20) | 0.197–4.563 | 18 | 0–180 | 19 |
| GEM Graupel (ID33) | 0.01–5.0 | 44 | N/A | 1 |

profiles of temperature and humidity from the nearest-in-time radiosonde were used as input to the D3R algorithm to calculate attenuation from atmospheric gases. These profiles are also used to qualitatively evaluate the retrieval output with respect to locations of well-known thermodynamic importance (e.g., the -15°C dendritic growth maximum and layers that are supersaturated with respect to ice).

## 3 Particle Scattering Properties

Gaining useful information from polarimetric radar measurements requires scattering properties of hydrometeors with preferred mean orientations. We incorporate such scattering properties into the retrieval algorithm described herein, by way of lookup tables (LUTs) that are derived by integrating these scattering properties over a prescribed particle size distribution (PSD). Specifically, we use scattering properties for a pristine plate, an aggregate of plates, and graupel (all at a wide range of sizes) from the Atmospheric Radiative Transfer Simulator (ARTS; Eriksson et al., 2018; Brath et al., 2020) database. Scattering properties for each of these particles are available for a discrete set of frequencies (for this study, we use the calculations at 13.4 GHz and 35.6 GHz), temperatures (190K, 230K, and 270K), incident angles of the transmitted radiation, scattering angles of the scattered radiation, and, except for graupel (which is assumed to have total random orientation) zenith-relative orientation angles (we refer to this angle hereafter as $\beta$). Table 2 lists values of these properties that correspond to the scattering calculations.

The ARTS database aims to provide scattering properties for a set of particles over a large range of frequencies so that applications using both active and passive remote sensing measurements are consistent. The derivation of polarimetric radar variables from this database is described in Appendix A. For each particle type listed in Table 2, these variables are integrated over a particle size distribution (PSD) using a modified gamma form (e.g., Petty and Huang, 2011):

$$N(D) = N_0 D^\mu exp[-\frac{\mu+4}{D_m}D], \tag{1}$$

where $D$ is the diameter of an equivalent-volume solid ice sphere in mm, $D_m$ is the mass-weighted mean equi-volume diameter, and $\mu$ is the shape parameter. While this definition of $D$ (and $D_m$) complicates the comparison to in-situ measurements (where maximum dimension is typically used to describe size), it is directly related to particle mass, and, in the Rayleigh limit, radar reflectivity. To reduce the dimensionality of the LUTs while preserving the variables that control the shape of the PSD, only $D_m$ and $\mu$ are varied in the construction of the LUT, and $N_0(D_m, \mu)$ is a normalized concentration factor such that the ice



**Table 3.** Dimensions of the lookup tables derived from the particle types selected from the ARTS Single Scattering Database.

| Dimension | Range |
|---|---|
| Frequency (f) | 3–94 GHz |
| Temperature (T) | 190–270 K |
| Zenith Angle ($\theta$) | 0–180° |
| Mean mass-weighted equivolume diameter ($D_m$) | 0.03–2.5 mm |
| PSD shape parameter ($\mu$) | -1–5 |
| Orientation dispersion parameter ($\kappa$) | 0–25 |

water content of all PSDs stored in the LUT is 1 g m$^{-3}$:

$$N_0(D_m, \mu) = \frac{1 \text{ g m}^{-3}}{\int_{D_{min}}^{D_{max}} \frac{\pi}{6} \rho_i D^{3+\mu} N \exp[-\frac{\mu+4}{D_m}] dD}, \tag{2}$$

where $\rho_i$ is the density of solid ice (0.917 g cm$^{-3}$). The distribution is later scaled by a retrieved concentration factor to match the observed reflectivity. For the particle types with preferred orientation, these PSDs are further integrated over the range of $\beta$ angles to account for the orientation distribution. We choose the von Mises distribution to represent the pdf of beta angles (Table 2). The von Mises distribution is a continuous function that represents the dispersion of variables in circular coordinates, and has been used to represent hydrometeor orientation retrieved from polarimetric radar (Bringi and Chandrasekar, 2001; Melnikov and Straka, 2013). For a zero mean canting angle, this distribution simplifies to:

$$p(\beta) = \frac{e^{\kappa cos\beta}}{G(\kappa)}, \tag{3}$$

where $\kappa$ is a dispersion parameter and $G(\kappa)$ is a normalization factor such that the sum of probabilities is equal to one. When $\kappa = 0$, the distribution is uniform; as $\kappa$ increases, the distribution becomes narrower and can be approximated by a normal distribution with standard deviation $\sqrt{1/\kappa}$. In the construction of the LUTs, we make the simplifying assumption that $\kappa$ is independent of particle size. While this is almost certainly an oversimplification (observations and Reynolds number analysis suggest that smaller particles of a given aspect ratio should have more randomly distributed canting angles; Klett, 1995), in the retrieval this is dealt with by allowing the combination of pristine and aggregate PSDs of different $\kappa$ values, as will be further described in Section 4.

The LUTs are constructed for each particle species. The LUT dimensions and ranges of the LUT indices are given in Table 3. The variables stored within the LUTs encompass three categories: radar variables, physical variables, and simulated PIP measurements. The radar variables are the single-particle scattering properties necessary to construct the polarimetric radar measurements: the reflectivity factor $Z$ at both horizontal and vertical polarizations, the extinction coefficient $k$ at both polarizations, the specific differential phase $K_{dp}$, and the real and imaginary parts of the copolar conjugate product of scattering





**Table 4.** Variables stored within the lookup tables.

| Group | Variables | Dimensions |
|---|---|---|
| Radar Variables | $Z_{vv}, Z_{hh}, k_{ext,v}, k_{ext,h}, K_{dp},$ | $f, T, \theta, D_m, \mu, \kappa$ |
| | $\mathrm{Re}(S_{hh}), \mathrm{Re}(S_{vv}), \mathrm{Im}(S_{hh}), \mathrm{Im}(S_{vv})$ | |
| Physical Variables and | $S, S_V, \rho_P, N_P^*$ | $D_m, \mu, \kappa$ |
| Simulated PIP measurements | $D_{mP}, \overline{a_{box}}, \overline{a_{ell}}$ | |

amplitudes ($C_{hv} = S_{hh}^* S_{vv}$). The formulas used to forward-model the radar measurements from these quantities are given in
Appendix B.

In addition to the radar variables, we simulate several variables (denoted with subscript $P$) that represent PIP-measured quantities. Because the PIP measures 2D projections of 3D particles, assumptions must be made to convert the PIP measurements to physical variables. While several formulas have been proposed to achieve this purpose (e.g., Jiang et al., 2017), we opted instead to use 2D projections of particles derived from the shapefiles used in the ARTS database to simulate the PIP
measurements over the range of $D_m$, $\mu$, and $\kappa$ in the LUTs, as this process is less ambiguous than the alternative of attempting to derive the 3D properties from the 2D PIP measurements. Moreover, this process ensures an internally consistent comparison of the radar retrieval output with PIP measurements. Most of these quantities depend in some way on the equivalent diameter $D_{eq}$ measured by the PIP. The simulated PIP measurements include the snowfall water equivalent rate ($S$), volumetric snowfall rate ($S_V$), effective density ($\rho_P$), normalized intercept $N_P^*$, area-weighted mean particle volume diameter $D_P$, mean box as-
pect ratio ($\overline{a_{box}}$) and mean ellipse aspect ratio ($\overline{a_{ell}}$). The final two quantities are weighted by the projected area. The snowfall water equivalent rate is calculated as:

$$S = 3.6 \int\limits_{\beta} p(\beta) \int\limits_{D_{min}}^{D_{max}} N(D)m(D)V_t(D,\beta)dDd\beta, \tag{4}$$

where $m(D)$ is the mass of the particle in grams, $V_t(D,\beta)$ is the terminal velocity in m s$^{-1}$, and S is in units of mm hr$^{-1}$. The terminal velocity is calculated following the process given in Heymsfield and Westbrook (2010), where the area ratio
is simulated from the ARTS shapefiles (and thus depends on both $D$ and $\beta$) and density/viscosity of air typical at the MHS location during ICE-POP 2018 snow events (-5°C, 90%RH, 925 hPa). The volumetric snowfall rate is calculated using the same terminal velocity, but, instead of mass, using the volume of the particle derived from the PIP-measured diameter:

$$S_V = 3.6 \int\limits_{\beta} p(\beta) \int\limits_{D_{min}}^{D_{max}} N(D) \frac{\pi}{6} D_{eq}^3(D,\beta) V_t(D,\beta) dDd\beta. \tag{5}$$

The PIP-derived density ($\rho_P$) is defined as the ratio of the liquid-equivalent snow rate to the volumetric snowfall rate multiplied
by the density of liquid water $\rho_w$ (Tiira et al., 2016):

$$\rho_P = \rho_w \frac{S}{S_P}. \tag{6}$$





The normalized intercept $N_P^*$ is adapted from the definition given for $N_{0,23}^*$ in Field et al. (2005):

$$N_P^* = \frac{\left[\int_\beta p(\beta) \int_{D_{min}}^{D_{max}} N(D) d_{max2D}^2(D,\beta) dD d\beta\right]^4}{\left[\int_\beta p(\beta) \int_{D_{min}}^{D_{max}} N(D) d_{max2D}^3(D,\beta) dD d\beta\right]^3}, \tag{7}$$

where $d_{max2D}(D,\beta)$ is the average 2D projected maximum particle dimension. This parameter is useful for providing temperature-dependent constraints on the PSD as will be shown in section 4.

$D_P$ is the ratio of the 4th to 3rd moments of the PSD in terms of $D_{eq}$:

$$D_P = \frac{\int_\beta p(\beta) \int_{D_{min}}^{D_{max}} N(D) D_{eq}^4(D,\beta) dD d\beta}{\int_\beta p(\beta) \int_{D_{min}}^{D_{max}} N(D) D_{eq}^3(D,\beta) dD d\beta}. \tag{8}$$

In order to evaluate the capability of the D3R polarimetric measurements to retrieve a bulk measurement of the aspect ratio, we defined the area-weighted mean ellipse aspect ratio ($\overline{a_{ell}}$):

$$\overline{a_{ell}} = \frac{\int_\beta p(\beta) \int_{D_{min}}^{D_{max}} N(D) D_{eq}^2(D,\beta) \frac{b}{a}(D,\beta) dD d\beta}{\int_\beta p(\beta) \int_{D_{min}}^{D_{max}} N(D) D_{eq}^2(D,\beta) dD d\beta}, \tag{9}$$

where $\frac{b}{a}(D,\beta)$ is the ratio of the short to long axis of the ellipse fitted to the simulated PIP image. In order to compare our retrievals to a variable that is influenced by the canting angle distribution parameter $\kappa$, we defined the area-weighted mean box aspect ratio ($\overline{a_{box}}$):

$$\overline{a_{box}} = \frac{\int_\beta p(\beta) \int_{D_{min}}^{D_{max}} N(D) D_{eq}^2(D,\beta) \frac{y}{x}(D,\beta) dD d\beta}{\int_\beta p(\beta) \int_{D_{min}}^{D_{max}} N(D) D_{eq}^2(D,\beta) dD d\beta}, \tag{10}$$

where $\frac{y}{x}(D,\beta)$ is the ratio of the vertical to horizontal dimensions of the bounding box of the simulated PIP image.

While it is not feasible to illustrate every dimension of the LUTs, a few examples are provided to preview the expected sensitivity of the D3R measurements to microphysical parameters. In Figure 2, the Ku-Ka dual-wavelength ratio is plotted as a function of both the volume-equivalent $D_m$ and PIP-measured $D_P$ for each of the three species. The DWR is nearly identical for the pristine plates and plate aggregates at $D_m < 1mm$ and $D_P < 3mm$, but reaches an upper limit of about 5 dBZ for the single plates while increasing to nearly 10 dBZ for the aggregates. Large Ku-Ka DWRs, sometimes exceeding 10 dBZ, have been observed coincident with large aggregates (20mm projected diameter) in dendritic growth regimes during OLYMPEX (Chase et al., 2018), so it is important that the LUTs capture this DWR range. The DWR of graupel is lower than the unrimed plates and aggregates for a given volume-equivalent size, but is similar or greater when viewed with respect to PIP-measured size based on the projected area. This suggests that DWR alone is not sufficient to determine the PSD mean particle size and density. Such information can be provided by the polarimetric measurements. For this study, we assume graupel to be randomly oriented and thus have zero contribution to $Z_{dr}$ and $K_{dp}$, which is a reasonable assumption for dry graupel (Kumjian, 2013). Meanwhile, the contributions to $Z_{dr}$ and $K_{dp}$ from plates and aggregates primarily depend on $D_m$ and $\kappa$ (Figure 3). As



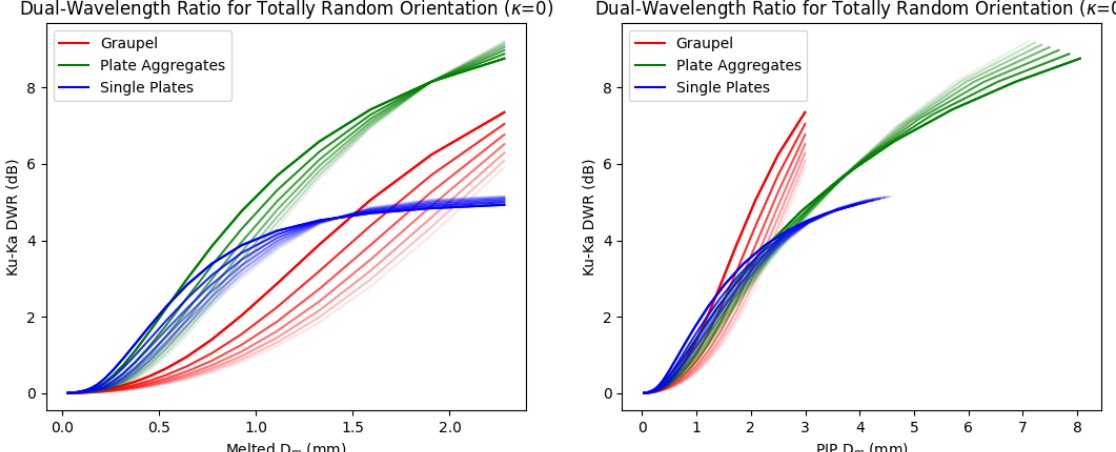

**Figure 2.** Ku-Ka dual-wavelength ratio for the three ice species listed in Table 2 as a function of the melted (volume equivalent) $D_m$ (left) and $D_P$ (right). The families of curves represent different values for the shape parameter $\mu$ ranging from -1 to 5, with lighter shades indicating higher values and less dispersion of the PSD. The different ranges of $D_P$ are a consequence of the different size-density relationships for each species.

expected, both $Z_{dr}$ and $K_{dp}$ increase with $\kappa$ as the orientation distribution becomes less dispersive. The $Z_{dr}$ rapidly decreases with aggregate size, as does $K_{dp}$ (though less rapidly). Meanwhile, the $Z_{dr}$ of the individual plates does not depend on $D_m$,

since these plates have a fixed aspect ratio at sizes larger than about 0.2 mm (Brath et al., 2020). The $K_{dp}$ of the individual plates also does not depend much on size, although some resonance effects lead to a small decrease in $K_{dp}$ at larger size. From Figures 2 and 3, it is clear that there is complementary information in the dual-frequency and polarimetric variables to discern particle size and species, which will be demonstrated in the next section.

## 4   Algorithm Description

Optimal estimation (OE; Rodgers, 2000) is a form of Bayesian inversion that assumes Gaussian error statistics and accommodates moderately nonlinear forward models. OE has been used for single-frequency (L'Ecuyer and Stephens, 2002; Munchak and Kummerow, 2011) and multi-frequency (Grecu et al., 2011; Mason et al., 2018) radar precipitation retrievals. It is applied here to the multi-frequency, polarimetric observations provided by the D3R. In this section, the OE components (state vector, observation vector, and covariance matrices) are defined and the approach is illustrated with an example retrieval along a single

radar ray.



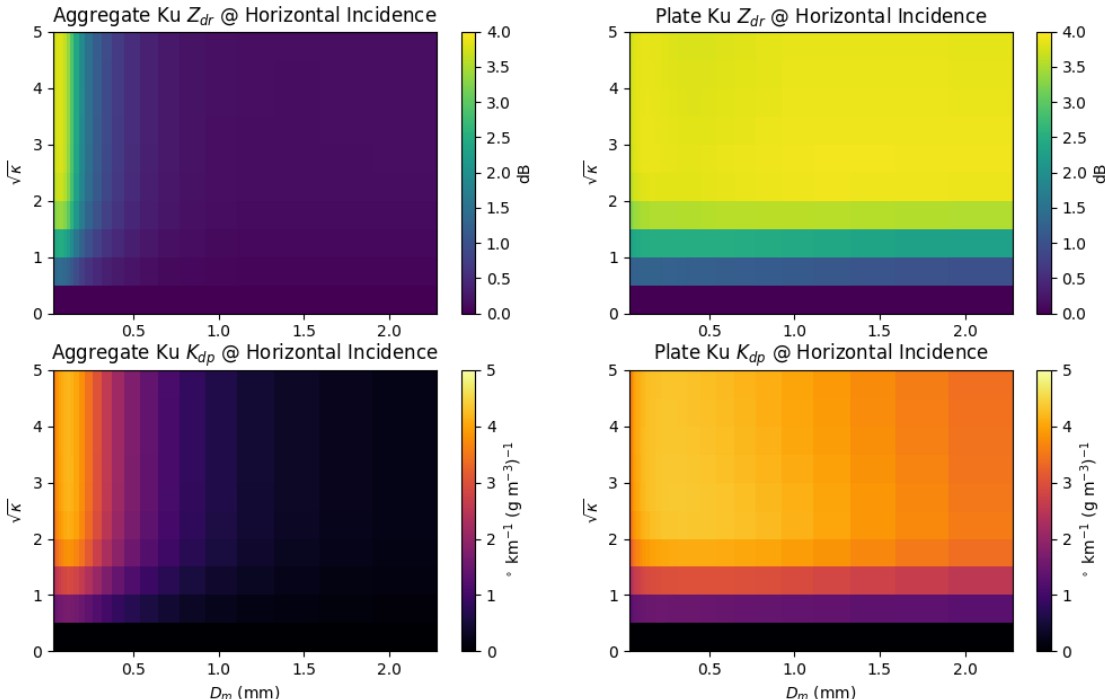

**Figure 3.** Polarimetric variables $Z_{dr}$ (top) and $K_{dp}$ (bottom) as a function of mean particle size $D_m$ and orientation distribution parameter $\kappa$. The left column represents aggregates, and the right column is for individual plates. All values are for horizontal incidence at Ku band at 270 K.

## 4.1 Optimal estimation setup

The cost function that is minimized by optimal estimation is

$$\Phi = \left(\mathbf{Y_{sim}}(\mathbf{X}) - \mathbf{Y_{obs}}\right)^{\mathbf{T}} \mathbf{S_y}^{-1} \left(\mathbf{Y_{sim}}(\mathbf{X}) - \mathbf{Y_{obs}}\right) + \left(\mathbf{X} - \mathbf{X_a}\right)^{\mathbf{T}} \mathbf{S_a}^{-1} \left(\mathbf{X} - \mathbf{X_a}\right), \tag{11}$$

where $\mathbf{Y_{obs}}$ is the observation vector and $\mathbf{Y_{sim}}(\mathbf{X})$ its forward-modeled counterpart, $\mathbf{S_y}$ is the measurement and forward model error covariance matrix, $\mathbf{X}$ is the state vector and $\mathbf{X_a}$ its prior, and $\mathbf{S_a}$ is the state covariance matrix. For the D3R retrieval of snow microphysical properties, we define $\mathbf{Y_{obs}}$ to contain one or more of the following series of dual-frequency or





dual-polarization observations at each range gate along a radial:

$$
\mathbf{Y_{obs}} = \begin{bmatrix}
\mathrm{DWR}^1 \cdots \mathrm{DWR}^{nDWR} \\
Z_{drKu}^1 \cdots Z_{drKu}^{nZdrKu} \\
Z_{drKa}^1 \cdots Z_{drKa}^{nZdrKa} \\
\phi_{dpKu}^1 \cdots \phi_{dpKu}^{n\phi dpKu} \\
\phi_{dpKa}^1 \cdots \phi_{dpKa}^{n\phi dpKa}
\end{bmatrix},
\tag{12}
$$

where each type of observation is filtered for ground clutter and only considered if the signal-to-noise ratio exceeds 1. Thus,
the number of observations of each type may differ. Note that $\phi_{dp}$ instead of $K_{dp}$ was chosen because $\phi_{dp}$ is the more direct measurement, and additional assumptions (with potentially non-Gaussian errors) are required to derive $K_{dp}$ from noisy $\phi_{dp}$ measurements.

The state vector consists of quantities that describe the PSD, relative contribution of each species, as well as other quantities known to affect the Ku- and Ka-band polarimetric radar measurements:

$$
\quad \mathbf{X} = \begin{bmatrix}
\delta N_P^{*1} \cdots \delta N_P^{*nNP} \\
f_p^1 \cdots f_p^{nfp} \\
f_r^1 \cdots f_r^{nfr} \\
D_p^1 \cdots D_p^{nDp} \\
\kappa_p^1 \cdots \kappa_p^{n\kappa p} \\
\kappa_a^1 \cdots \kappa_a^{n\kappa a} \\
\mu_p^1 \cdots \mu_p^{n\mu p} \\
\mu_a^1 \cdots \mu_a^{n\mu a} \\
c^1 \cdots c^{nc}
\end{bmatrix},
\tag{13}
$$

where each quantity is defined at nodes (indicated by the superscript in Eq. 13) which may be arbitrarily placed (for this study, nodes are spaced 600m horizontally and 250m vertically). Each of these quantities, their priors, and ranges are defined in Table 5. Following Grecu et al. (2011), the measured Ku-band $Z_{hh}$ (which is *not* in $\mathbf{Y_{obs}}$) is used as direct input to the forward model, and from these measurements and the quantities in the state vector $\mathbf{X}$, the measurements in $\mathbf{Y_{obs}}$ are simulated. A
detailed description of this forward model is provided in Appendix B.

The observation error covariance matrix $\mathbf{S_y}$ must accurately describe the error characteristics of the measurements and forward model. Values that are unrealistically low can lead to overfitting, whereas overly conservative (large) error estimates can lead to under-utilization of the information contained within the measurements. In this study we consider the diagonal elements of $\mathbf{S_y}$ to be the sum of the measurement error and forward model error components, which are given in Table 6 for
each measurement in $\mathbf{S_y}$. The measurement errors are obtained from the gate-to-gate variance over many homogeneous scenes at low ($<10°$) elevation angles, where the assumption is that the true change in the measured quantity is small compared to the measurement noise. The measurement error is assumed to be uncorrelated in space and between variables. The forward model error is quantified by assessing the variance in the measurements for alternate aggregate particles of the same size as





**Table 5.** Names, definitions, *a priori* value, standard deviation, and allowed ranges of quantities in the state vector (Eq. 13). Note that some quantities are retrieved in logarithmic space to accomodate lognormal distributions and/or to linearize the forward model.

| Symbol | Definition | Prior | Std. Dev. | Minimum | Maximum |
|---|---|---|---|---|---|
| $\delta N_P^*$ | Deviation of $\log_{10}(N_P^*)$ from expected value given by Field et al. (2005): $N_P^* = 5.65e5\exp[-0.107*(T_C)]$ | 0 | 1.5 | -3 | 3 |
| $f_p$ | Fraction of total mass contributed by pristine particle model | 0.5 | 0.5 | 0.01 | 0.99 |
| $f_r$ | Fraction of aggregate mass contributed by rimed particle model | 0 | 0.5 | 0 | 0.99 |
| $D_p$ | Ratio of $D_m$ of the pristine particle population to $D_m$ of the aggregate particle population | 0.5 | 0.5 | 0.1 | 1 |
| $\kappa_p$ | Orientation distribution parameter $\ln(\kappa + 0.1)$ of the pristine particle population | 0 | 2 | $\ln(0.1)$ | $\ln(25)$ |
| $\kappa_a$ | Orientation distribution parameter $\ln(\kappa + 0.1)$ of the aggregate particle population | 0 | 2 | $\ln(0.1)$ | $\ln(25)$ |
| $\mu_p$ | Shape parameter of the size distribution of pristine particles | 2 | 2 | -1 | 5 |
| $\mu_a$ | Shape parameter of the size distribution of aggregate particles | 2 | 2 | -1 | 5 |
| $c$ | $\log_{10}$(Cloud liquid water in g m$^{-3}$) | -2 | 2 | -5 | 0 |

**Table 6.** Measurement and forward model components of the error covariance matrix $\mathbf{S_y}$ for each measurement, expressed as standard deviations.

| Measurement | Measurement error | Forward Model Error | Total |
|---|---|---|---|
| $DWR$ | 1.91 dB | 0.76 dB | 2.05 dB |
| $Z_{drKu}$ | 0.46 dB | 0.80 dB | 0.93 dB |
| $Z_{drKa}$ | 0.46 dB | 0.80 dB | 0.93 dB |
| $\phi_{dpKu}$ | 1.85° | 43% | $\sqrt{(1.85°)^2 + 0.43\phi_{dpKu}^2}$ |
| $\phi_{dpKa}$ | 1.86° | 43% | $\sqrt{(1.86°)^2 + 0.43\phi_{dpKa}^2}$ |

the ARTS aggregates. These alternate models include dendrites and columns, with different assumptions about the aggregation process (Schrom et al., 2021). Although these forward model errors are correlated between variables, analysis of the covariance matrices showed the correlation between $DWR$ and $Z_{dr}$ error to be insignificant. While $Z_{dr} - K_{dp}$ error correlation is larger, the $\phi_{dp}$ error is dominated by upstream propagation error which is assumed to be uncorrelated to the $Z_{dr}$ forward model error at a given range gate. Therefore, for this study the off-diagonal elements of $\mathbf{S_y}$ are set to zero, although this is a choice that could be refined in future implementations of the retrieval.



## 4.2 Example ray

The optimal estimation process along a ray of radar data is illustrated in Figure 4, which shows the observed ($\mathbf{Y_{obs}}$) and simulated ($\mathbf{Y_{sim}}$) measurements, and Figure 5, which shows the various retrieval parameters in $\mathbf{X}$ as well as derived quantities of ice water content and $D_m$ for each iteration until convergence. This ray is characterized by a DWR peaks at 0-5 km and 23-28 km, which reach 6 dB, dropping to 1-3 dB elsewhere. The $Z_{dr}$ has several peaks, the most significant of which reach over 2 dB at 20 km and 30 km range, bracketing the DWR peak. The $\phi_{dp}$ increases most rapidly in the 10-20 km range with smaller rates of increase elsewhere, reaching 15° at Ku band and 50° at Ka band. The initial profiles of the retrieval parameters $N_P^*$, $f_r$, $f_p$, $\kappa_p$, $\kappa_a$, $\mu_p$, $\mu_a$, and the derived ice and cloud liquid water content and $D_m$ of the aggregate and pristine PSDs are shown in the lightest shaded lines in Figure 5. The iterative adjustments guided by the Jacobian matrix respond to the initial differences between $\mathbf{Y_{obs}}$ and $\mathbf{Y_{sim}}$:

- $N_P^*$ decreases slightly near the DWR peaks and increases elsewhere;

- $f_p$ decreases below 50% in the DWR peak region, but increases above 60% in the 10-20 km range and again near 30 km, corresponding to the $Z_{dr}$ peaks and steepest increase in $\phi_{dp}$;

- $f_r$ is generally below 40%, with minima near the $Z_{dr}$ peaks;

- $\kappa_p$ is generally lower than $\kappa_a$, but exhibits peaks corresponding to the $Z_{dr}$ peaks, whereas $\kappa_a$ does not vary as much with range;

- $\mu_a$ and $\mu_p$ increase slightly from their initial values in most regions, although $\mu_a$ dips below zero near the DWR peaks (lower $\mu$ corresponds to a longer tail of the PSD at large sizes and can result in higher DWRs, all else being equal);

- No significant cloud liquid water is detected or needed to explain the DWR observations. In fact, the near-zero DWR at the far ranges imply that there is little differential attenuation, and the observed near-field DWR can be attributed to particle size effects.

- $D_m$ of the pristine population (controlled by the retrieval parameter $D_p$) generally increases as a proportion of the $D_m$ of the aggregates in order to balance pristine particle contributions to $Z_{dr}$ and $\phi_{dp}$.

The most significant impact of these parameter changes is to increase $D_m$ and reduce ice water content in the DWR peak regions, with the opposite changes elsewhere. The final retrieved state is in good agreement with the $Z_{dr}$ and $\phi_{dp}$ observations at both frequencies. There is a residual high bias in the DWR, especially at the range beyond 30 km. Examining the DWR histogram for this case in Figure 1 reveals a mean near or below zero, which may indicate a low bias in the relative calibration of the Ku to Ka reflectivity, and possible high bias in the ice water content retrieval.

## 4.3 Information Content Analysis

Some further insight into the adjustments made to the parameters can be gained by examining the Jacobian matrix $\mathbf{K}$ of partial derivatives of each element of $\mathbf{Y_{sim}}$ with respect to each element of $\mathbf{X}$:





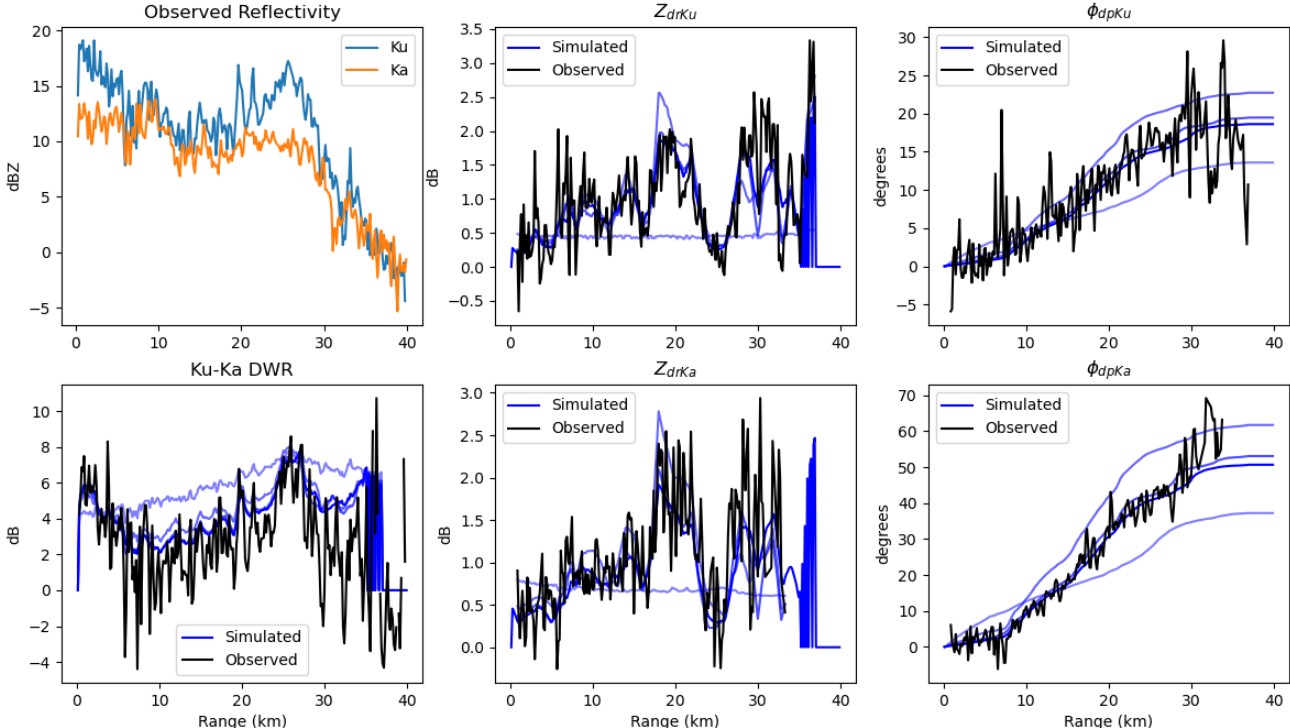

**Figure 4.** D3R observations of reflectivity (upper left) and DWR (lower left), $Z_{dr}$ (middle), and $\phi_{dp}$ (right) along the $231°$ azimuth, $6°$ elevation radial at 0217 UTC 8 March 2018. For the DWR, $Z_{dr}$, and $\phi_{dp}$, the D3R observations are indicated by the solid black lines and the simulated measurements for each iteration are shown in progressively more opaque blue lines.

While the Jacobian is state-dependent, the sign and relative magnitude of each element from the example ray shown in the previous section is illustrative of the general sensitivity of the forward model to the retrieval parameters. The Jacobian matrix is composed of blocks that are either diagonal or triangular, depending on whether the parameter has a significant downrange propagation effect on an observation. The effect of modifying the parameters in **X** can be explained by considering the change

to the PSD at a fixed Ku-band reflectivity:

– Increasing $N_P^*$ results in a smaller mean particle size and higher ice water content. This decreases the DWR and increases $\phi_{dp}$ downrange, while the effect on $Z_{dr}$ is relatively small and state-dependent.

– Increasing $f_p$ increases $Z_{dr}$ and $\phi_{dp}$ downrange as a result of the increasing contribution of the pristine particles to the PSD, with little effect on DWR.

– Increasing $D_p$ also increases $Z_{dr}$ (and decreases DWR) as the pristine particles become larger and contribute more to reflectivity, but decreases $\phi_{dp}$ downrange as the ice water content in reduced.





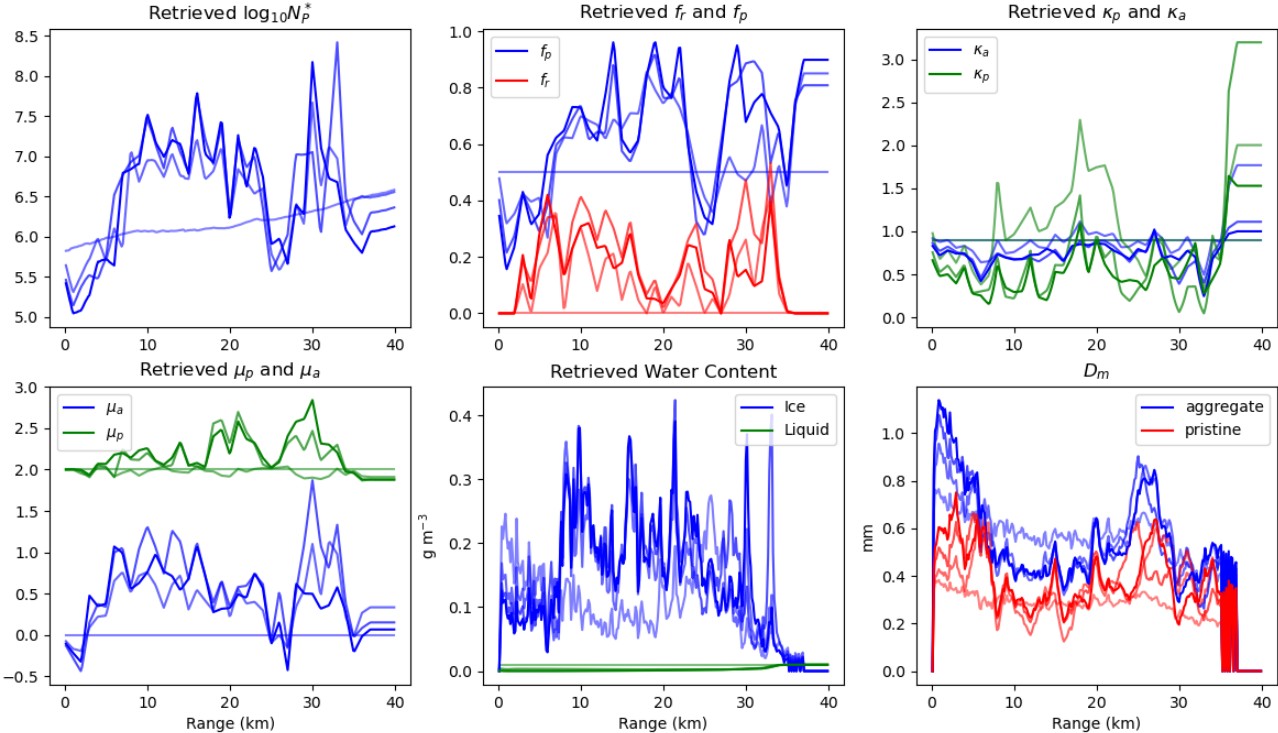

**Figure 5.** D3R observations of reflectivity (upper left) and DWR (lower left), $Z_{dr}$ (middle), and $\phi_{dp}$ (right) along the $231°$ azimuth, $10°$ elevation radial at 0130 UTC 8 March 2018. For the DWR, $Z_{dr}$, and $\phi_{dp}$, the D3R observations are indicated by the solid black lines and the simulated measurements for each iteration are shown in progressively more opaque blue lines.

- Increasing $\kappa_a$ and $\kappa_p$ increases the $Z_{dr}$ and downrange $\phi_{dp}$, with the $\kappa_p$ having a much larger magnitude effect.

- Increasing $\mu_a$ reduces the DWR as the large-particle tail of the PSD is truncated. There is almost no impact of changing $\mu_p$ on the simulated observations.

- Increasing cloud liquid results in an increase in the downrange DWR due to increased differential attenuation, but no impact on the polarimetric measurements.

- Increasing $f_r$ reduces the DWR and all of the polarimetric measurements, as the rimed particles are assumed to have random orientation.

It is noteworthy that the sign of the observation response to perturbations varies in different ways for the different parameters in **X**. This is an indication that this optimal estimation retrieval, as we have formulated it, is well-determined and is also a requirement for reducing ambiguity, or cross-talk, in the retrieved state. The information content and cross-talk among



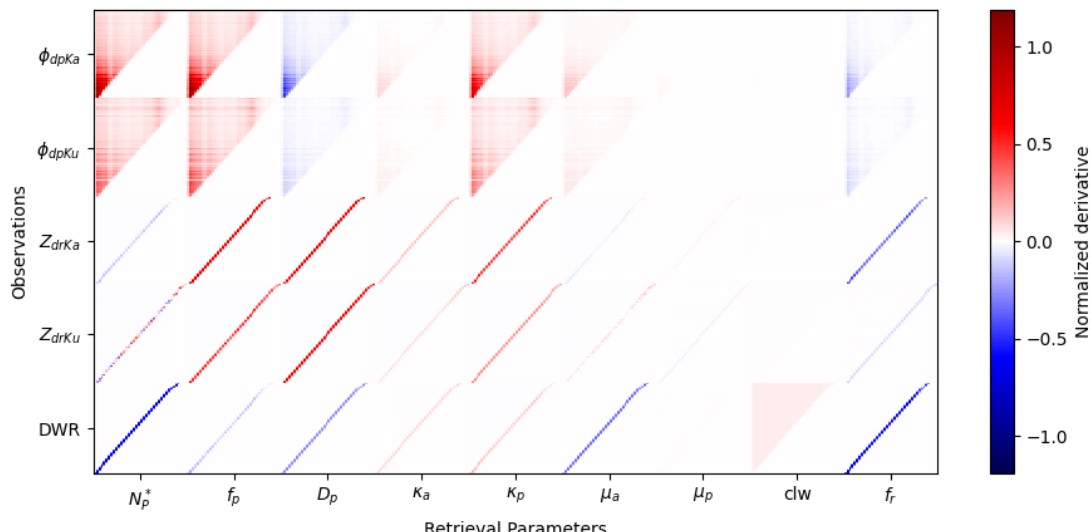

**Figure 6.** Jacobian matrix of partial derivatives of the simulated D3R measurements to the retrieval parameters. The partial derivatives have been normalized element-wise by the square root corresponding diagonal element of $\mathbf{S_y}$ – *i.e.*, the expected standard deviation of that measurement's error.

parameters can also be evaluated by examining the averaging kernel matrix of the retrieved state. The averaging kernel provides a measure of influence of the observations on the retrieved state, and is defined as:

$$\mathbf{A} = \left(\mathbf{S_a}^{-1} + \mathbf{K^T S_y}^{-1}\mathbf{K}\right)^{-1}\left(\mathbf{K^T S_y}^{-1}\mathbf{K}\right). \tag{4}$$

Values close to one indicate strong influence of measurements, and values close to zero indicate that the retrieval is heavily influenced by the prior. In Figure 7, the median, 10%, and 90% quantiles of the averaging kernel are plotted. These statistics were derived from many retrieved states spanning the cases we examined in Table 1. The parameter with the highest information content is the normalized intercept ($N_P^*$), which both the DWR and $\phi_{dp}$ are highly sensitive to (Figure 6). The two parameters that describe the pristine component of the PSD ($f_p$ and $D_p$) also have a similar median and 90% quantile values to $N_P^*$, owing

to their sensitivity to the polarimetric parameters, but a lower 10% quantile, likely originating from situation where the $f_r$ is high and there is little sensitivity of the polarimetric variables to $f_p$ and $D_p$. Similarly, $\kappa_p$ has a large range between the 10% and 90% quantile indicating that the information content of this parameter is state-dependent, and high values of $f_p$ and $D_p$ are required to maximize the sensitivity of the polarimetric observations to this variable. Another parameter with a wide range of averaging kernel diagonal values is $f_r$, which requires low $f_p$ and $D_p$ values for it to be the primary driver of the polarimetric

variables. Some of this state dependence of the information content is also reflected in the off-diagonal values, which indicate significant cross-talk, *i.e.*, a strong correlation in the posterior state vector, between the following groups of variables: ($N_P^*$,





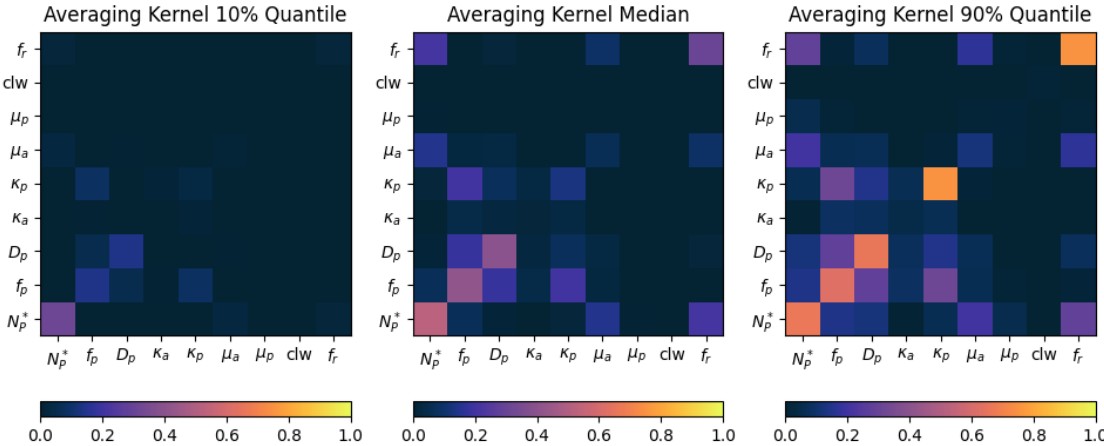

**Figure 7.** Quantiles of the averaging kernel matrix diagonal elements and off-diagonal elements representing different parameters at the same range gate.

$\mu_a$, $f_r$) and ($f_p$, $D_p$, $\kappa_p$). These groups have similar Jacobians making it difficult to determine them independently. Finally, it is notable that a few of the variables ($\kappa_a$, $\mu_p$, and cloud liquid) have very low averaging kernel values, indicating that the observations are not particularly sensitive to them. This is not surprising, since the aggregates do not show much $Z_{dr}$ or $K_{dp}$

sensitivity to $\kappa$ except at the smallest sizes (Figure 3, and the shape parameter ($\mu$) of the pristine PSD is not going to influence the reflectivity observations (DWR and $Z_{dr}$ because the shape of the large-particle tail is only important to these observations if $D_p$ is close to its upper limit of one. The sensitivity of $Z_{dr}$ and $K_{dp}$ to $\kappa$ does depend on the aggregate shapes. The ARTS large plate aggregate used herein may not represent cases of more horizontally exaggerated aggregates where the assumed orientation will have a larger impact on the simulated polarimetric variables. The low averaging kernel values for cloud liquid

are reflective of the result that it was rarely retrieved in significant quantities, and since it is treated logarithmically, only large values will induce large changes to the DWR. However, there were several RHI scans, particularly on February 28, where some cloud liquid was necessary to explain high DWR values that could not be achieved by differential scattering alone.

## 5   Validation

The primary tool used for validation of the D3R retrievals is the PIP located at the MHS site. The D3R retrievals were matched

to the PIP by averaging the retrieved quantities in a 600 m wide by 500 m tall box centered above MHS. The lower altitude limit of this box was placed 250 m above the surface to avoid ground clutter contamination along the radials used for averaging. To evaluate the impact of the dual-frequency and dual-polarization measurements, four retrieval experiments were conducted:





- – Ku-only: A single-frequency retrieval, equivalent to prescribing a temperature-dependent $Z - S$ relationship;

- – DWR-only: A dual-frequency retrieval without polarimetric information; similar information content to the GPM Dual-Frequency Precipitation Radar (DPR);

- – Ku-pol: A single-frequency polarimetric retrieval using $Z_{dr}$ and $\phi_{dp}$ at Ku-band only;

- – All-obs: A dual frequency polarimetric retrieval using DWR and the $Z_{dr}$ and $\phi_{dp}$ at both frequencies.

To assess the quality of the PIP-D3R matchups, the Ku-band reflectivity was calculated directly from the PIP-derived PSDs using Mie theory (spherical particles) with the PIP-derived particle densities (Tokay et al., 2021). Although there will be some departures from Mie theory for non-spherical particles, at Ku band these are relatively small (Kuo et al., 2016). The larger contribution to error is the various assumptions required to derive particle density from the PIP size and fall speed (Tokay et al., 2021). Using an ensemble approach to these assumptions, a range of reflectivities was obtained and compared to the D3R-observed reflectivity in the averaging box (top row of Figure 8).

Some different tendencies are noted for each event. The 9 January case had a consistently higher PIP-derived reflectivity than the D3R measurement, especially during the middle hours of the event. This was a low-echo-top cold low case, and MRR radar profiles indicate significant echo growth (perhaps due to aggregation) at low levels, which may have continued in the clutter region. The 28 Feb case exhibits a similar bias between the PIP-derived and D3R-measured reflectivity at Ku band in the first 6 hours, after which the D3R reflectivity comes within the lower range of PIP estimates. This was a deeper, warm low case, and the MRR and D3R profiles do not indicate any substantial reflectivity increase towards the surface. Instead, it is suspected that excess attenuation from wet snow (the wet bulb temperature was above $0°C$ until 0400 UTC and the air temperature was above $0°C$ until 0800 UTC) accumulated on the radome is responsible for these differences (note also the sharp increase in DWR during the same time period attributed to this factor in section 2). The 7–8 Mar case exhibited the best agreement between PIP-derived and D3R-measured reflectivity throughout the event. This was also a warm low case with deep echo tops, but colder wet bulb temperatures (between $-4°C$ and $-2°C$ during the event).

The validation statistics presented in this section are derived from matchups of the D3R-derived and PIP-measured quantities listed in Table 4. Because the PIP measurements are taken every minute, whereas the D3R RHI scans were conducted every 5-6 minutes, an optimal lag was found by maximizing the correlation between the lagged D3R-measured and PIP-derived Ku-band reflectivity time series. This lag time was between 2-7 minutes, depending on event, which is consistent with a fall speed slightly greater than 1 m/s to cover the distance from the center of the averaging box to the surface. Only retrievals where the D3R-measured Ku band reflectivity was within the range of PIP estimates were considered in the calculation of statistics to ensure that the PIP measurements are reasonably representative of the D3R observations.

## 5.1 Snowfall rate and water equivalent

The time series of snow water equivalent rate ($S$) for each event are shown in Figure 9. The D3R substantially underestimates snowfall throughout the 9 Jan and 28 Feb events, which is not surprising since the PIP-derived reflectivity significantly exceeds





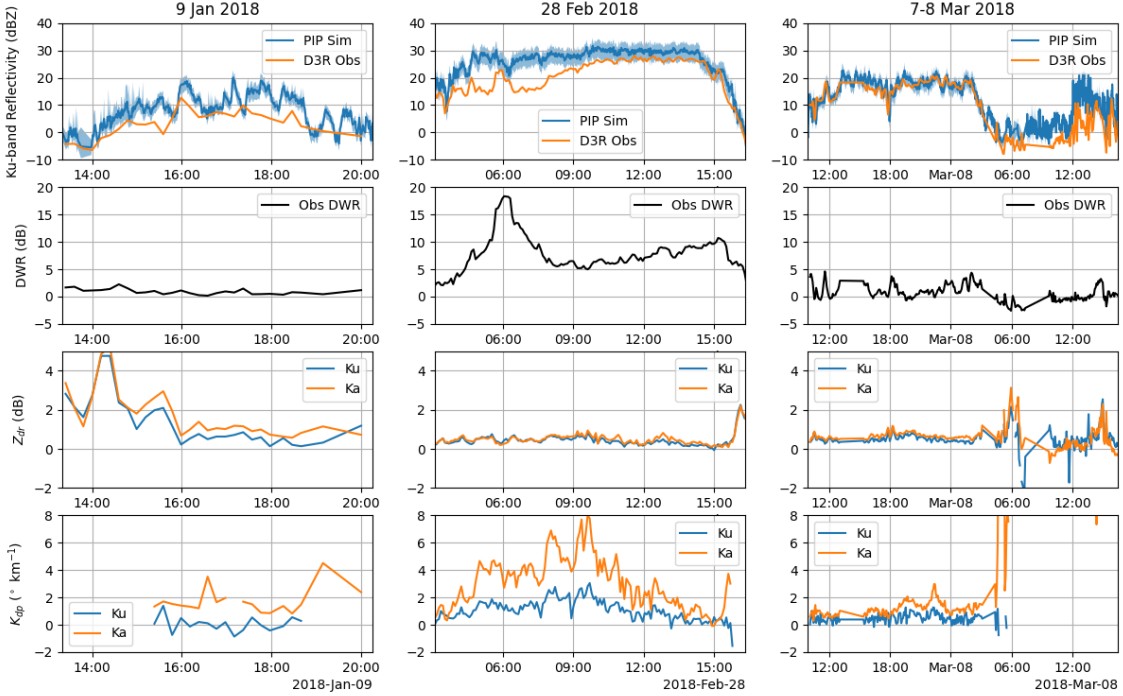

**Figure 8.** Time series of D3R-observed and PIP-simulated Ku-band reflectivity (top row), D3R-observed DWR (second row), D3R-observed $Z_{dr}$ (thrid row), and D3R-derived $K_{dp}$ for each event. The PIP simulations used Mie calculations for spherical particles with the PIP-derived effective density. Because these calculations involve a variety of assumptions to derive density from particle shape and fall speed (Tokay et al., 2021), a range between the minimum and maximum from these calculations is shaded around the mean PIP-derived value.

the D3R measurement for reasons discussed previously in this section. It is worth noting, however, that the DWR-only and All-Obs retrievals are in good agreement with the Pluvio-measured $S$ accumulation on 9 Jan, which would be consistent with aggregation processes in the lowest levels that increase reflectivity, but do not increase $S$. The 7–8 Mar event shows better agreement with the PIP measurements. In this case, which had the best reflectivity match, the DWR-only retrieval overestimates $S$ whereas the Ku-only and Ku-pol retrievals underestimate $S$ relative to the PIP. The retrieval using all of the observations

is the closest match and the accumulated $S$ falls within the range of PIP estimates for most of the event. We note from the DWR time series for this event in Figure 8 that the DWR is just above 0 dB for much of this event which implies a smaller mean particle size and higher $S$ for a given reflectivity. Meanwhile, the Ku-pol retrieval gives a slight reduction in $S$ from the Ku-only retrieval, which is already biased low for this event. Compared to the Pluvio, the All-obs retrieval is biased high, however, after correcting the Pluvio data for wind (Milewska et al., 2019), this retrieval is in better agreement, however, these

same wind corrections bias the 9 Jan event higher than the retrievals and in better agreement with the PIP.





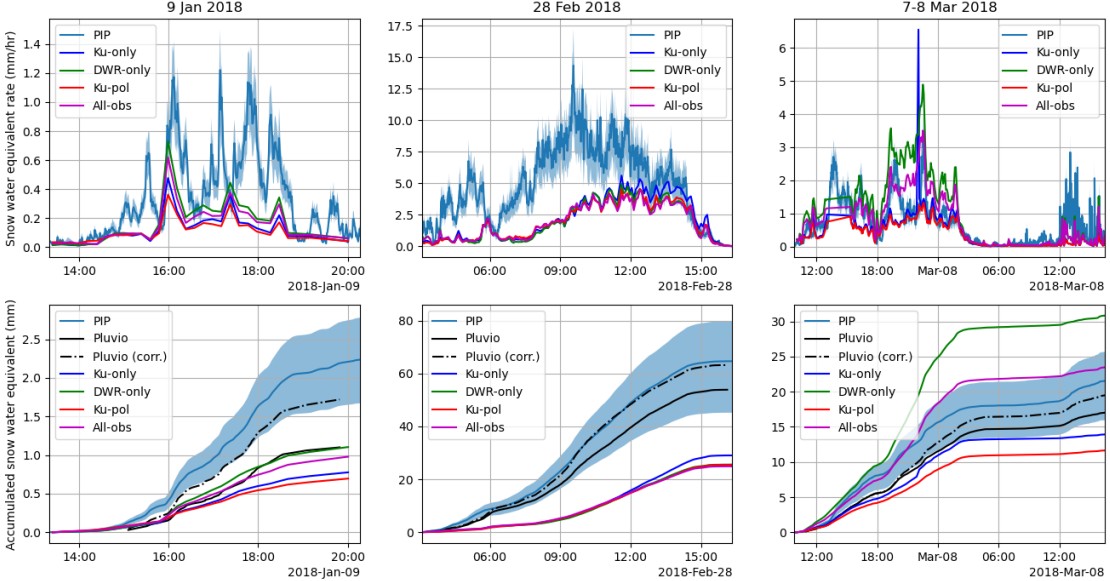

**Figure 9.** Time series of D3R- and PIP-derived snow water equivalent rate (top) and accumulation (bottom) for each event. Because the PIP calculations involve a variety of assumptions to derive particle mass from shape and fall speed (Tokay et al., 2021), a range between the minimum and maximum from these calculations is shaded around the mean PIP-derived value.

Error statistics for volumetric snowfall rate ($S_V$) and $S$ from all events, filtered for times when the D3R-measured reflectivity was within the PIP-estimated range, are presented in Table 7. The bias is the overall fraction difference between the accumulated PIP- and D3R-derived amounts. All methods underestimate the amounts, with the DWR-only method providing the closest match for both $S_V$ and $S$. However, this appears to be the result of compensating biases on the 28 Feb and 7–8 Mar events, and the MAE is highest for this method. The Ku-pol method provides the best correlation for $S_V$ and $S$ even though it has the largest magnitude bias of all the experiments.

## 5.2 Mean particle size and density

The D3R retrieval provides an estimate of the PIP-measured area-weighted mean particle volume diameter ($D_P$) that is consistent with the particle shapes used to generate the LUTs and can be compared directly to the PIP measurement. The time series of this parameter is shown for each event in the top row of Figure 10. On 9 Jan 2018, all of the radar retrievals were biased low with respect to the PIP, which is consistent with the reflectivity bias we noted on this day. On 28 Feb, the PIP measured a rapid increase of $D_P$ to a maximum at 06 UTC, followed by a decrease to a minimum around 09 UTC and another maximum at 15 UTC. None of the retrievals did a particularly good job of capturing the peak at 06 UTC, but all methods except the Ku-only retrieval captured the increase toward the second maximum, although again, the peak values were underestimated. On 7–8 Mar,





**Table 7.** Mean bias, absolute error (MAE), and correlation coefficient ($r$) of PIP-derived versus D3R-estimated volumetric snowfall rate ($S_V$) and snow water equivalent rate ($S$) for each of the four retrieval experiments, compiled over all three events when the D3R-observed Ku-band reflectivity was within the PIP-simulated Ku-band reflectivity range. Because of the wide dynamic range of snowfall rates, these quantities are expressed as percentages relative to the mean PIP estimate.

| Experiment | $S_V$ Bias (%) | $S_V$ MAE (%) | $S_V$ $r$ | $S$ Bias (%) | $S$ MAE (%) | $S$ $r$ |
|---|---|---|---|---|---|---|
| Ku-only | -31.2 | 39.8 | 0.891 | -29.7 | 42.4 | 0.872 |
| DWR-only | -30.8 | 39.5 | 0.941 | -7.9 | 77.6 | 0.789 |
| Ku-pol | -46.9 | 42.4 | 0.955 | -41.9 | 40.5 | 0.918 |
| All-obs | -38.3 | 40.0 | 0.948 | -24.1 | 52.7 | 0.860 |

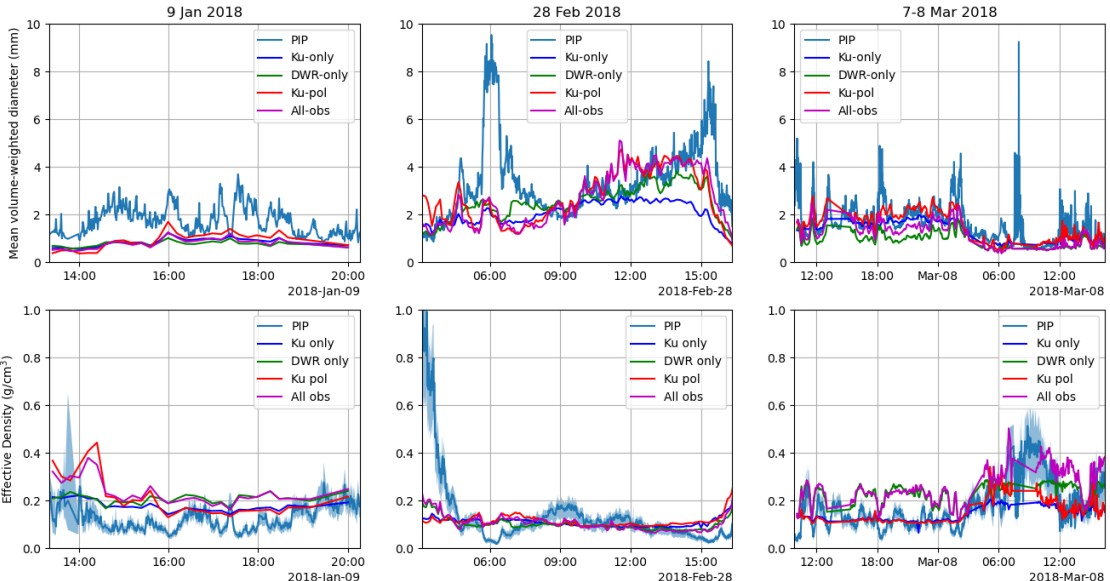

**Figure 10.** Time series of D3R- and PIP-derived mean volume-weighted particle diameter (top) and effective density (bottom) for each event. Because the PIP calculations involve a variety of assumptions to derive particle density from shape and fall speed (Tokay et al., 2021), a range between the minimum and maximum from these calculations is shaded around the mean PIP-derived value.

both methods that used the DWR (DWR-only and All-obs) captured the temporal variability of $D_P$ quite well, but were biased low; this is consistent with a suspected low bias in the DWR for this event (mean cloud-top DWRs were slightly below zero; see Figure 1).

The statistics of the $D_P$ retrievals are presented in Table 8 and, as with the snowfall rate statistics, only consider observations where the D3R-observed reflectivity was within the range of calculated PIP values. All of the retrievals are biased low with





**Table 8.** Mean bias, absolute error (MAE), and correlation coefficient ($r$) of PIP-derived versus D3R-estimated mean volume-weighted particle diameter ($D_v P$) and PIP effective density ($\rho_P = S/S_V$) for each of the four retrieval experiments.

| Experiment | $D_P$ Bias (mm) | $D_P$ MAE (mm) | $D_P$ $r$ | $\rho_P$ Bias (g cm$^{-3}$) | $\rho_P$ MAE (g cm$^{-3}$) | $\rho_P$ $r$ |
|---|---|---|---|---|---|---|
| Ku-only | -0.74 | 0.84 | 0.590 | -0.02 | 0.05 | 0.459 |
| DWR-only | -0.89 | 0.93 | 0.736 | 0.05 | 0.08 | 0.393 |
| Ku-pol | -0.13 | 0.68 | 0.610 | -0.01 | 0.05 | 0.363 |
| All-obs | -0.44 | 0.70 | 0.709 | 0.05 | 0.08 | 0.428 |

respect to the PIP, with the Ku-pol retrieval coming the closest. This is counter-intuitive, since this retrieval does not consider the DWR, which is the measurement most sensitive to $D_P$. However, the suspected low bias of the DWR on 7-8 Mar, which dominates the statistics due to its close match to the observed Ku reflectivity, contributes heavily here. The MAE is only slightly lower for the Ku-pol method than the All-obs method, despite the much more significant bias. Meanwhile, the correlation coefficient is largest for the two methods that incorporate the DWR, suggesting that the DWR is indeed informative regarding

the particle size; this underscores the need for DWR to be carefully calibrated to avoid significant bias. Combined use of polarimetric and DWR information seems to at least partially alleviate these biases.

The retrieved effective particle density $\rho_P$, defined as the snow water equivalent rate divided by the volumetric snowfall rate, can also be directly compared to the PIP measurement. To first order, $\rho_P$ is inversely proportional to $D_P$ because the unrimed aggregates' density decreases with size. However, changes in $f_r$ and $f_p$ also affect $\rho_P$ so it is worthwhile to evaluate

these retrievals as well. The bottom row of Figure 10 shows the time series of $\rho_P$ for each event. In general, we find that when the retrieved $D_P$ is biased high with respect to the PIP observation, $\rho_P$ is biased low and vice-versa. It is interesting that on 28 Feb, all of the D3R methods are in tight agreement whereas on 7-8 Mar, the DWR-only and All-obs retrievals are higher than the Ku-only and Ku-pol retrievals, due to the shift toward smaller (denser particles) inferred from the low DWR on this event. In this case, the Ku-only and Ku-pol methods are less biased, but the variability is better represented by the DWR-using

retrievals. This is again borne out in the statistics (Table 8), where the Ku-only and Ku-pol methods have the lowest magnitude bias and MAE, and unlike the $D_P$, the Ku-only method has the best correlation (although the All-obs retrieval is the next best). This suggests that the size-density relationship in the scattering models we chose may not be representative of the particles observed during ICE-POP, or that more information (e.g., W-band reflectivity) is needed to constrain the density (e.g., Kneifel et al., 2015).

**5.3  Bulk particle orientation and aspect ratio**

The use of polarimetric measurements in the Ku-pol and All Obs retrievals has the most impact on the retrieved pristine fraction, ratio of pristine to aggregate mean particle size, and pristine population orientation distribution. All of these parameters combine to influence the area-weighted ellipse ($\overline{a_{ell}}$) and box ($\overline{a_{box}}$) aspect ratios. In Figure 11, the influence of the $Z_{dr}$ and $K_{dp}$ measurements (Figure 8) can be observed in these two retrievals, whereas the Ku-only and DWR-only experiments did





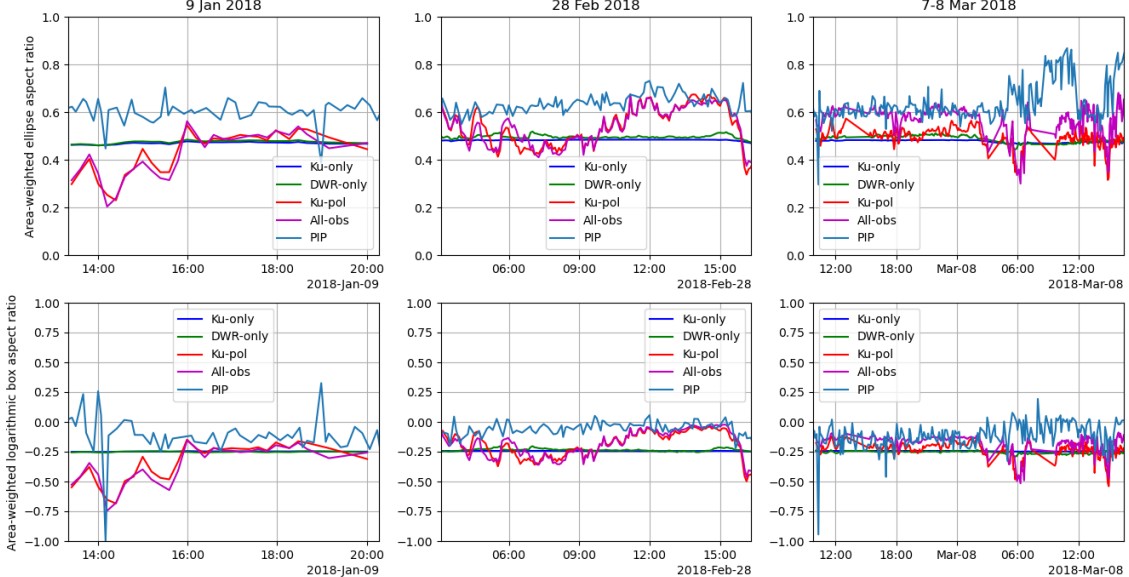

**Figure 11.** Time series of D3R- and PIP-derived area-weighted ellipse (top) and box (bottom) aspect ratio for each event.

not appreciably change these parameters. The high $Z_{dr}$ values before 16 UTC on 9 Jan result in low aspect ratios at that time. There is not much of a trend in $Z_{dr}$ on 28 Feb, owing to the constant presence of large aggregates which dominate the $Z_{dr}$ measurement, but there is a notable peak in $K_{dp}$ around 09 UTC which corresponds to the minimum aspect ratio for this event. The 7–8 Mar event had the least variable polarimetric signatures, but a short peak in $Z_{dr}$ around 18 UTC on 7 Mar corresponds to a drop in aspect ratio at that time.

The comparison of the retrieved aspect ratios to those derived from the PIP is inconsistent from event to event. There appears to be little variability in the PIP time series on 9 Jan for either measure of aspect ratio. There is a steady increase in $\overline{a_{ell}}$ between 09 and 12 UTC on 28 Feb, matching the retrieved behavior, although the PIP range is considerably smaller than the retrieved range. The 7-8 Mar event does not show any significant trend in the PIP-measured aspect ratios, consistent with the low $Z_{dr}$ variability during this event. The aspect ratio error statistics are given in Table 9. The Ku-pol and All-obs methods provide

the lowest MAE and least bias, however, the best correlation comes from the Ku-only method (for $\overline{a_{ell}}$) and the DWR-only method (for $\overline{a_{box}}$). This is a surprising result, especially in light of the very small range of these retrieved values in Figure 11. In these methods, the very limited AR variability is entirely driven by changes in PSD, with smaller mean particle sizes being associated with smaller aspect ratios. The polarimetric measurements add significant variation to the retrieved aspect ratio, but the correlation statistics suggest that the relationship between these measurement and PIP-derived aspect ratio and orientation

is tenuous at best.



**Table 9.** Mean bias, absolute error (MAE), and correlation coefficient ($r$) of PIP-derived versus D3R-estimated mean area-weighted ellipse ($\overline{a_{ell}}$) and box ($\overline{a_{box}}$) aspect ratio for each of the four retrieval experiments.

| Experiment | $\overline{a_{ell}}$ Bias | $\overline{a_{ell}}$ MAE (mm) | $\overline{a_{ell}}$ $r$ | $\overline{a_{box}}$ Bias | $\overline{a_{box}}$ MAE | $\overline{a_{box}}$ $r$ |
|---|---|---|---|---|---|---|
| Ku-only | -0.14 | 0.14 | 0.383 | -0.66 | 0.66 | 0.406 |
| DWR-only | -0.12 | 0.12 | 0.134 | -0.66 | 0.66 | 0.431 |
| Ku-pol | -0.10 | 0.10 | 0.373 | -0.59 | 0.59 | 0.281 |
| All-obs | -0.06 | 0.07 | 0.138 | -0.55 | 0.55 | 0.180 |

## 6 Conclusions

This study describes an algorithm that makes use of both polarimetric and dual-frequency radar measurements to retrieve microphysical properties of falling snow, including: snowfall rate (volumetric and water equivalent); ice water content; particle size distribution; the relative contribution of pristine, aggregate, and rimed species; and particle orientation distribution. The

algorithm is flexible in that it can use as many or as few measurements as available. In this study, it is applied to the Ku- and Ka-band measurements of the NASA D3R radar obtained during the ICE-POP 2018 field campaign, but can be applied to additional or different frequencies. This is possible because it makes use of the ARTS microwave single scattering property database for oriented particles (Brath et al., 2020), which encompasses ADDA scattering calculations over a wide range of frequencies. This differentiates it from methods that use T-Matrix or Rayleigh-Gans approximations, but constrains it to use

the particle geometries that are available (at this time, only hexagonal plates and aggregates composed of these plates). More geometries are available for randomly-oriented particles, but these can not make use of the polarimetric information (although they are used in this study to represent the rimed particles).

The retrieval uses optimal estimation to solve for several parameters that describe the PSD, relative contribution of each species, and the orientation distribution along an entire radial simultaneously. This is necessary (versus a gate-by-gate ap-

proach) to account for the measurements sensitive to propagation effects (*e.g.*, $DWR$ and $\phi_{dp}$). Examination of Jacobian matrices and averaging kernels show that the DWR provides information regarding the characteristic particle size, and to a lesser extent, the rime fraction and shape parameter of the size distribution. The $Z_{dr}$ measurements provide information regarding the mass fraction of pristine particles and their characteristic size and orientation distribution. Meanwhile, the $\phi_{dp}$ measurements are sensitive to most of the same measurements as $Z_{dr}$, but are also sensitive to the overall particle concentration. Thus, by

combining the dual-frequency and polarimetric measurements, some ambiguities can be resolved that should allow a better determination of the particle size distribution parameters and integrated quantities (*e.g.*, ice water content, snowfall rate) than can be retrieved from single-frequency polarimetric measurements or dual-frequency, single-polarization measurements.

The D3R ICE-POP retrievals were validated using PIP and Pluvio measurements taken nearby at the May Hills ground site. The PIP measures the snow PSD directly (Pettersen et al., 2020) and several useful parameters can be derived directly from

its measurements or indirectly with additional assumptions. These include the snowfall rate (volumetric and water equivalent),





mean volume-weighted particle size, and effective density (Tokay et al., 2021), as well as parameters describing the mean aspect ratio and orientation distribution. We validated the retrieval during 3 events representing both warm and cold snow regimes Kim et al. (2021). These events were chosen based upon availability of both PIP and D3R data, significant accumulation at the MHS site, and absence of any mixed-phase precipitation which the algorithm does not account for. Four retrieval experiments

were performed to evaluate the utility of different measurement combinations: Ku-only, DWR-only, Ku-pol, and All-obs. In terms of mean absolute error and correlation, the volumetric snowfall rate was best-retrieved ($r = 0.95$), followed closely by the snow water equivalent rate ($r = 0.92$). The Ku-pol method had the highest correlation to these parameters while the DWR-only and All-obs methods had the lowest magnitude bias. These methods that incorporated DWR also had the best correlation to particle size ($r = 0.74$), although none of the methods retrieved density particularly well ($r = 0.46$). The ability of the

measurements to retrieve mean aspect ratio was also inconclusive, although the polarimetric methods (Ku-pol and All-obs) had reduced biases and MAE relative to the Ku-only and DWR-only methods. The significant biases in particle size and snowfall rate appeared to be related to biases in the measured DWR (positive on 28 Feb and negative on 7-8 Mar), emphasizing the need for accurate DWR measurements and frequent calibration (e.g., co-located measurements at a non-attenuating frequency such as S- or C-band). Notwithstanding these calibration biases, during the most well-behaved event (7-8 Mar), where the

PIP-derived reflectivity was closest to the D3R measurement, the All-Obs method provided the best snowfall accumulation and closely approximated the observed time series of snowfall rate and particle size.

The D3R is scheduled to be deployed in Storrs, Connecticut, USA during the 2021-22 and 2022-23 winters as part of the NASA-sponsored Investigation of Microphysics and Precipitation for Atlantic Coast-Threatening Snowstorms (IMPACTS) field experiment. A similar deployment setup is planned with nearby PIP measurements. Additionally, the airborne Ku- and

Ka-band HIWRAP radar will be available to evaluate the D3R calibration, and other lessons learned (e.g., video monitoring to observe snow accumulation on the radome, siting to avoid ground clutter) will be applied. Ongoing improvements to the PIP processing algorithms, particularly regarding the estimation of particle aspect ratio, will also be advantageous to further refine the algorithm described in this work. Availability of scattering databases for oriented particles with different geometries will facilitate running these retrievals as an ensemble to provide more robust posterior distributions of the retrieved parameters.

Finally, the methodology can be expanded to accommodate liquid and melting particles, although scattering databases for the latter, particularly with the polarimetric parameters from oriented melting particles, are not yet mature enough for this application.

*Data availability.* Chandrasekar, V. . 2019. GPM Ground Validation Dual-frequency Dual-polarized Doppler Radar (D3R) ICE POP. Dataset available online from the NASA Global Hydrology Resource Center DAAC, Huntsville, Alabama, U.S.A. DOI:

http://dx.doi.org/10.5067/GPMGV/ICEPOP/D3R/DATA101

Bliven, Larry . 2020. GPM Ground Validation Precipitation Imaging Package (PIP) ICE POP. Dataset available online from the NASA Global Hydrology Resource Center DAAC, Huntsville, Alabama, U.S.A. DOI:

http://dx.doi.org/10.5067/GPMGV/ICEPOP/PIP/DATA101



## Appendix A:  Derivation of single-scattering properties

Faithfully simulating the observables from polarimetric radar requires considering the incident and scattered Stokes vectors; these vectors are related via (Adams and Bettenhausen, 2012)

$$
\begin{bmatrix} I_s \\ Q_s \\ U_s \\ V_s \end{bmatrix} = \frac{1}{r^2} \begin{bmatrix} Z_{11} & Z_{12} & Z_{13} & Z_{14} \\ Z_{21} & Z_{22} & Z_{23} & Z_{24} \\ Z_{31} & Z_{32} & Z_{33} & Z_{34} \\ Z_{41} & Z_{42} & Z_{43} & Z_{44} \end{bmatrix} \begin{bmatrix} I_i \\ Q_i \\ U_i \\ V_{i,} \end{bmatrix}
\tag{A1}
$$

where $I$, $Q$, $U$, and $V$ are the elements of the Stokes vector, $r$ is the distance from the sensor to the particle, $Z_{lm}$ are the elements of the scattering or phase matrix, and the $i$ and $s$ subscripts indicate incidence and scattering, respectively.

To generate tables of the polarimetric, single-scattering properties, we transform the Stokes matrix elements to single-scattering properties more commonly used in radar meteorology, such as backscatter cross-section at horizontal and vertical polarizations ($\sigma_{hh}^b$ and $\sigma_{vv}^b$, respectively). Additionally, we include the complex, co-polar conjugate product $C_{hv}$ between the scattering amplitude matrix elements at horizontal and vertical polarization ($S_{hh}$ and $S_{vv}$, respectively) that allow for the co-polar correlation coefficient $\rho_{hv}$ to be calculated. These variables are defined in terms of the phase matrix elements as (Ekelund

et al., 2020)

$$
\sigma_{hh}^b = 2\pi(Z_{11} + Z_{12} + Z_{21} + Z_{22})
\tag{A2}
$$

$$
\sigma_{vv}^b = 2\pi(Z_{11} - Z_{12} - Z_{21} + Z_{22})
\tag{A3}
$$

$$
Re(C_{hv}) = \frac{Z_{44} + Z_{33}}{2}
\tag{A4}
$$

$$
Im(C_{hv}) = \frac{Z_{43} - Z_{34}}{2}.
\tag{A5}
$$

All the expressions above define the phase matrix elements as in the backscatter direction, or opposite the incident direction, and the phase matrix elements have units of mm$^2$.

Expressions are given below for the radar reflectivity factor

$$
Z_{h,v} = \frac{\lambda^4}{\pi^5 |K_w|^2} \int\limits_0^\infty \int\limits_0^\pi N(D)p(\beta)\sigma_{h,v}(D,\beta)d\beta dD
\tag{A6}
$$





in units of $\mathrm{mm}^6\mathrm{m}^{-3}$ and co-polar correlation coefficient

$$\rho_{hv} = \frac{4\lambda^4}{\pi^4 |K_w|^2} \frac{\int_0^\infty \int_0^\pi N(D)p(\beta)\sqrt{Re(C_{hv}(D,\beta))^2 + Im(C_{hv}(D,\beta))^2}d\beta dD}{\sqrt{Z_h Z_v}}. \tag{A7}$$

The propagation variables, $K_{dp}$ and the extinction cross-sections ($\sigma_h^e$ and $\sigma_v^e$) can be calculated from the corresponding extinction matrix of the particle. The oriented, azimuthally random uniform particles we use from the ARTS database have only three independent extinction matrix elements: $K_{11}$, $K_{12}$, and $K_{34}$. The propagation variables are thus defined as (Ekelund

et al., 2020)

$$\sigma_h^e = K_{11} + K_{12} \tag{A8}$$

$$\sigma_v^e = K_{11} - K_{12} \tag{A9}$$

$$K_{dp} = \frac{180 \times 10^{-3}}{\pi} \int\limits_0^\infty \int\limits_0^\pi N(D)p(\beta)K_{34}(D,\beta)d\beta dD. \tag{A10}$$

The extinction matrix elements have units of $\mathrm{mm}^s$ and $K_{dp}$ has units of $\mathrm{^\circ km^{-1}}$.

## Appendix B:  Radar Forward Model

The D3R measurements that comprise the observation vector $\mathbf{Y}$ (Eq. 12) are forward modeled from the measured Ku-band vertically-polarized reflectivity $Z_{v,Ku}^m$[1] and the state vector $\mathbf{X}$. This combination of frequency and polarization was chosen

because it is least prone to error in the attenuation correction. To simulate the D3R measurements, we must obtain both the backscattering properties of the ice particles as well as the propagation scattering properties (i.e., the propagation phase shifts and attenuation at each polarization) from the lookup tables (LUTs) defined in Section 3. The components of $\mathbf{Y}$ are defined by the following equations that include the propagation effects along a radial:

$$\mathrm{DWR}(r) = 10\log_{10}Z_{v,Ku}^m(r) - 10\log_{10}Z_{v,Ka}^m(r), \tag{B1}$$

$$Z_{dr}(r,f) = 10\log_{10}Z_{h,f}^m(r) - 10\log_{10}Z_{v,f}^m(r), \tag{B2}$$

$$\phi_{\mathrm{dp}}(r,f) = 2\int\limits_0^r K_{\mathrm{dp}}(r',f)dr' + \phi_{\mathrm{sys}}, \tag{B3}$$

---

[1]Reflectivity is defined in linear units in this appendix unless otherwise noted.





where $K_{\mathrm{dp}}(f)$ is the specific differential phase at frequency $f$, $\phi_{\mathrm{sys}}$ is the system differential phase upon transmission, and $Z_{p,f}^m$ is the attenuated reflectivity at polarization $p$ and frequency $f$ defined as

$$Z_{p,f}^m(r) = \exp[-2 \int_0^r k_{\mathrm{ext}}(p,f,r')dr']Z_{p,f}^e(r). \tag{B4}$$

Here $Z_{p,f}^e$ is the intrinsic effective reflectivity factor and $k_{\mathrm{ext}}(p,f)$ is the specific attenuation, which includes contributions from gases, cloud liquid water, and hydrometeors. The symbols $p, f$ indicate the polarization and frequency, respectively.

DWR, $Z_{dr}$, and $\phi_{dp}$ are forward simulated from equations B1-B4 by integrating numerically along each radial, following a conceptually similar process to that of Grecu et al. (2011). First, at each range gate, the attenuation contributions from gases (determined by the closest-in-time sounding from DGW) and cloud liquid (part of the retrieval state vector) are calculated.
Next, $Z_{v,Ku}^e$ is calculated by inserting the observed $Z_{v,Ku}^m$ and the calculated 2-way path integrated attenuation from previous range gates into Eq. B4. Then, $Z_{p,f}^e$, $k_{\mathrm{ext}}(p,f)$, and $K_{\mathrm{dp}}(f)$, along with the simulated PIP measurements in Table 4, are calculated from interpolated LUTs that have been combined for all the species according to the retrieval parameters and scaled to the observed $Z_{v,Ku}^m$. These quantities are then used to forward-model the measurements that comprise $\mathbf{Y}$ following equations B1-B4.

The process of simulating the radar measurements from combined LUTs can be described by considering a LUT for a variable $V$ selected from Table 4. First, the LUT dimensionality for each species is reduced by linear interpolation over the following parameters: the radial zenith angle $\theta$, temperature at the altitude of the range gate, and species-specific $\mu$ and $\kappa$ values interpolated from the nodes for these parameters. At this stage, the remaining LUT dimensions are $D_m$ and frequency (for the radar variables). Next, the unrimed aggregate ($V_u$) and rimed ($V_r$) LUTs are combined into a merged "aggregate" LUT ($V_a$) by
weighting each species according to the rime mass fraction interpolated to each range gate $r$:

$$V_a(D_m) = (1 - f_r(r))V_u(D_m) + f_r(r)V_r(D_m). \tag{B5}$$

Recall from Section 3 that the LUTs are normalized to contain a constant $1\,\mathrm{g\,m^{-3}}$ ice water content, there exists a corresponding $N_{P0}^*(D_m)$ value consistent with that water content. Because $N_P^*$ is prescribed at each range gate from the interpolated retrieval parameter $\delta N_P^*(r)$ (see definition in Table 5), the LUT for each species $s$ is re-scaled element-wise to be consistent with this
prescribed $N_P^*$:

$$V_s^x(D_m) = V_s(D_m)\frac{N_P^*(r)}{N_{P0}^*(D_m)}. \tag{B6}$$

The pristine and aggregate LUTs are then merged – individually for the scaled and unscaled LUTs – according to the interpolated retrieval parameters $f_p$ (pristine mass fraction) and $D_p$ (ratio of the pristine to aggregate $D_m$). To perform this merging, first the mean mass-weighted diameter dimension of the combined LUT ($D_{m,c}$) is calculated from the prescribed $f_p$, $D_p$, and
aggregate $D_{m,a}$:

$$D_{m,c} = f_p(r)D_p(r)D_{m,a} + (1 - f_p(r))D_{m,a}. \tag{B7}$$





Next, the variables in the pristine LUTs ($V_p$ and $V_p^x$) are interpolated to this new set of $D_{m,c}$ values and merged with the aggregate LUTs ($V_a$ and $V_a^x$) according to the prescribed mass fraction to create a combined LUT:

$$V_c(D_{m,c}) = f_p(r)V_p(D_p(r)D_{m,a}) + (1 - f_p(r))V_a(D_{m,a}), \tag{B8}$$


$$V_c^x(D_{m,c}) = f_p(r)V_p^x(D_p(r)D_{m,a}) + (1 - f_p(r))V_a^x(D_{m,a}). \tag{B9}$$

Now there exists a one-dimensional LUT for each parameter (and frequency for the radar variables) that is indexed by $D_m$ and is consistent with the prescribed PSD characteristics from the retrieval state vector. The retrieved value of $D_m(r)$ is that which produces the attenuation-corrected (from Eq. B4 over prior range gates) $Z_{v,Ku}^c(r)$ from the $N_P^*$-scaled $Z_{v,Ku}$ LUT.

From this retrieved $D_m(r)$, the ice water content $W(r)$ (in g m$^{-3}$) is simply the scaling factor that is required to produce this $Z_{v,Ku}^c(r)$ from the unscaled LUT. Once $D_m(r)$ and $W(r)$ are determined, all of the remaining LUT parameters, including those needed to simulate the measurements in $\mathbf{Y}$ as well as the simulated PIP measurements used for validation, can be readily calculated by interpolating the LUTs to $D_m(r)$ and scaling by $W(r)$. These parameters are saved to the output data structure, and the propagation equations are iteratively integrated to repeat the process at the next range gate.

*Author contributions.* Dr. Munchak led this research by developing the lookup tables and optimal estimation methodology, processing the radar data, and calculating validation statistics. Dr. Schrom assisted with the generation of lookup tables, polarimetric components of the radar forward model, and assessment of forward model error. Dr. Tokay processed the PIP and Pluvio measurements that were used for validation. Dr. Helms performed additional processing of the PIP data to derive the bulk aspect ratio and orientation parameters.

*Competing interests.* S. J. Munchak is a member of the editorial board of Atmospheric Measurement Techniques. Otherwise, the authors
declare that they have no conflict of interest.

*Acknowledgements.* Dr. Schrom's and Dr. Helms' work was supported by an appointment to the NASA Postdoctoral Program at NASA Goddard Space Flight Center, administered by Universities Space Research Association under contract with NASA. Dr. Munchak and Dr. Tokay were supported by the Global Precipitation Measurement (GPM) Ground Validation program. The authors would also like to acknowledge many insightful discussions about the D3R data quality and calibration procedures with Prof. V. Chandrasekhar at Colorado
State University. Finally, the authors would like to acknowledge Prof. GyuWon Lee of Kyungpook National University, Daegu, South Korea for stimulating ongoing scientific research from datasets collected during ICE-POP 2018.





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
