# Peer review of "Snow Microphysical Retrieval from the NASA D3R Radar During ICE-POP 2018"

_Atmospheric Measurement Techniques, 2021_

## Author Comment (AC1)

This is a very solid and well written paper describing the optimal estimation (OE) algorithm for retrieval of microphysical characteristics of ice using combined polarimetric and dual-frequency radar measurements. The algorithm was tested on the data collected during the ICE-POP 2018 experiment in Korea. Although certain microphysical features of ice / snow are well captured, an overall quality of the algorithm performance is quite modest which might be possibly or partially attributed to the instrumental biases of the radar measurements (and the dual-wavelength ratio DWR in particular).

We thank the reviewer for their helpful comments.

A fundamental question this study raises regards the feasibility of utilizing a very complicated and computationally intense OE methodology to solve multiparameter problems with large uncertainties in the state and observed vectors. I do not exclude that combining more simplistic retrieval methods with careful data quality control might be more efficient under such scenario.
Here is a list of more concrete comments and suggestions,

- The authors avoid using specific differential phase $K_{DP}$ in their formalism and resort to the total differential phase instead. Radial dependencies of $\Phi_{DP}$ in Fig. 4 and temporal plots of $K_{DP}$ in Fig. 8 (bottom panels) show that $K_{DP}$ can be quite reliably estimated at both Ku and Ka bands. $K_{DP}$ is very sensitive to a lower end of the particle spectrum and the results of this and similar studies indicate that $K_{DP}$ is strongly correlated with the total concentration of smaller-size ice. In other words, $K_{DP}$ has very strong informative content and is immune to attenuation, resonance scattering effects (even at Ka band), and radar miscalibration.

We agree that $K_{dp}$ is an informative radar observable that does provide constraints on the smaller particle sizes and total particle number concentration, and is less prone to calibration error than the other measurements. One reason we avoid using $K_{dp}$ is the uncertainty in estimating the quantity from profiles of differential phase. This uncertainty comes from the variety of methods to calculate $K_{dp}$ as well as the noise in the PhiDP field. Our method accounts for the propagation effects (both attenuation and accumulation of differential phase from non-zero $K_{dp}$) directly, so the information content provided by the differential phase shift is utilized without having to take the additional step of estimating the $K_{dp}$ field and its associated uncertainties (which are likely to be non-Gaussian, violating the OE formalism) . In other words, it is more difficult to constrain the independent $K_{dp}$ values at each range gate than the total phase shift, since errors in $K_{dp}$ will accumulate down range if there are systematic errors in PSD and orientation parameters.

- Since the D3R radar was able to do genuine RHIs during the ICE-POP experiment, would it be possible to display composite RHIs of Z, $Z_{DR}$, and $K_{DP}$ and generate vertical profiles of the radar variables (at Ku and Ka bands) over the PIP location in a height vs time format? This would give a better idea about the vertical microphysical structure of the storm and possible problems in the radar – PIP comparison which are mentioned in the manuscript such as enhanced vertical gradients of Z likely responsible for underestimation of snow rate and size.

This is a good suggestion and we have generated these plots. The strong near-surface vertical gradient in $Z_H$ is clearly evident in the 9 January case, suggesting low-level ice particle growth may be one reason retrievals produce lower snowfall rates and particle sizes than those found from the PIP. The strong peak in DWR (without similar patterns in the other variables) is also evident in the 28 February case. We have added these plots to the revised manuscript.

[Figure]

*Figure R1: Time-height cross-sections of RHI profiles of the 9 January 2018 (left column), 28 February 2018 (middle column), and 7-8 March 2018 (right column) cases. The profiles correspond to 10 km downradial of the D3R radar, approximately 8 km downradial of the PIP. We use the 10-km range so that the elevation angles are low enough for the polarimetric variables to be meaningful. From top to bottom, row 1 is the Ku-band $Z_H$, row 2 is the Ku-band $Z_{DR}$, row 3 is the Ku-band $K_{dp}$, row 4 is the Ku-band $\rho_{hv}$, and row 5 is the Ku-Ka Dual-wavelength ratio.*

Captions to Figs. 4 and 5 are the same. Thank you for noticing this oversight, we have fixed the caption to Figure 5.

Correct the reference the Ryzhkov et al. (2016) paper. It is not in press.

Thank you for noticing this error, we have fixed the reference.

---

## Author Comment (AC2)

Title: Snow Microphysical Retrieval from the NASA D3R Radar During ICE-POP 2018

Manuscript summary:

The manuscript introduces an algorithm retrieving the particle size distribution and various microphysical properties of snow from dual-wavelength dual-polarization radar via optimization estimation. These estimations include characteristic particle size, shape parameter of the size distribution, rime fraction, mean aspect ratio, orientation distribution and snowfall rate (volumetric and water equivalent). The results are verified with the ground instrument during ICE-POP 2018. Most of the comparisons indicate high biases but with high correlation coefficients.

The major concern of the algorithm is trying to estimate various parameters (Eq. 13, 9 parameters at each gate) at the same time, yet the information for retrieval is limited (EQ. 12, 5 measurements). The validation indicates high biases. The manuscript needs substantial revision. Major revision is recommended.

We thank the reviewer for their effort in reviewing this manuscript.

Major comments:

1. Considering the number of variables (nine) is higher than measurements (five), it's highly possible the final retrievals are mostly from the "background" value (e.g., Prior in Table 5). The manuscript did not compare the PPI (or RHI )simulated measurements from "Prior" and "last iteration". This is important to ensure the retrieval is not mostly from "background" term (2nd term of RHS in Eq. 11). Figure 4 is the only figure to show that the final simulated measurements from optimal estimation. Yet, a ray data is not convincing. Please do show the PPI or RHI simulated measurements for all of the experiments including Hu-only, DWR-only, Ku-pol and All-obs.

This is a good suggestion. We have added a comparison plot of reflectivity, differential reflectivity, dual-wavelength ratio, and differential phase to the manuscript, with a discussion of how faithfully these variables are simulated using each set of measurements.

[Figure]

*Figure R2: Comparisons of Ku-band Z (column 1; from left to right), Ku-band $Z_{DR}$ (column 2), Ku-Ka Dual-wavelength ratio (column 3), Ku-band $\Phi_{dp}$ (column 4), and Ka-band $\Phi_{dp}$ (column 5). The top row are the observed values from an RHI taken at 13:00 UTC on 28 February at an azimuth of 231 degrees. Simulated measurements from the retrievals are shown for Ku-Z only (row 2), Ku-polarimetric (row 3), Ku-Z and Ka-Z only (row 4), and all observations (row 5).*

We selected the RHI from 13 UTC on 28 Feb, since this was a timestep where the DWR calibration appeared to be reasonable, though perhaps still having some positive bias (see middle column of Fig. R1). The Ku-only algorithm (which essentially returns the a priori result) simulates $Z_{DR}$ that is too high, DWR that is close to the observed values (but without some small-scale features), and $\Phi_{dp}$ that is too high at both frequencies. The Ku-pol

algorithm is able to match the polarimetric measurements (including the Ka-band $K_{DP}$, even though it is not an input to the Ku-pol algorithm, because the Ku-band and Ka-band $K_{DP}$ are highly correlated), but still does not resolve the small-scale features observed in the DWR. The DWR-only algorithm is able to resolve these, but suffers from the same $Z_{DR}$ and $K_{DP}$ bias as the Ku-only results. The retrieval using all observations is best able to match these features, although for this case, it is clear that the polarimetric observations are providing the most information to adjust the microphysical parameters from their *a priori* values.

2. Since the observational information is limited, fixing the value of some parameters (e.g., axis-weighted ellipse ratio, riming fraction). Figure 10 and 11 indicate that the effective density and ellipse ration has little change compared to other variables. Has author considered to reduce retrieved variables?

This is a reasonable suggestion. The purpose of this study was to investigate sensitivities to all parameters, so we consider it a useful result to identify those parameters to which the measurements are not sensitive. So, we can fix these parameters to improve efficiency in future use of this algorithm.

3. Three cases are show for validation. Two of the cases (9 Jan. and 28 Feb.) has pronounced bias between radar measurement and PIP simulation (Fig. 8). Later, the retrieval bias can also be noticed among those two cases (Figs. 9 and 10). In terms of snowfall rate, author can consider perform more comprehensive validation by including more Pluvio data.

There are certainly errors in the PIP-retrieved quantities stemming from uncertainties in the microphysical assumptions of particle shape, fall behavior, etc. Comparisons between the observed and simulated PIP quantities do provide some useful ways of evaluating the shape and particle property retrievals purely from radar measurements. These results do show discrepancies between the radar-based retrievals of snowfall rate and particle shape compared to those derived from the PIP.. We have made our explanation for why these discrepancies occur clearer to the reader. Additionally, Pluvio measures accumulation quite accurately but does not have the temporal resolution necessary to validate rate on short timescales. It is used primarily as a tool to understand biases in the PIP estimation method.

4. Overall, the validation is rather disappointing. The bias and MAR are around +/- 30% in snowfall rate, nearly 40% in volume-weighted diameter (Dp), 40% in area-weighted ellipse and box aspect ratio. It's a good sign that the correlation coefficient is high. Moreover, there is no PPI or RHI retrieval to ensure that the retrievals are consistent in spatial domain.

We have added comparisons of the simulated and observed RHIs for one case (in response to the prior comment). Some of the discrepancy between the simulated and PIP-derived

particle properties is due to the natural presence of irregular ice particles that we don't incorporate into the retrievals. Owing to the PIP location being downradial of the D3R, the validation radar gates are 250-750m above the PIP. Therefore, some bias is to be expected when the ice particles continue to undergo microphysical growth processes as they fall towards the ground, as shown by substantial vertical gradients in the polarimetric radar variables during the 9 Jan case (see Fig. R1). Additional, some radar calibration errors produce biases in the reflectivity and/or DWR that can in turn bias the retrievals. We agree that the high correlation coefficient is a good sign, since it indicates a response of the radar observations to the parameters the PIP is measuring, but that there is a bias in either the forward model (i.e., our selected particle scattering properties), the radar observations, or both. These sources of error in the validation are further discussed in the revised section 5.

Minor comments:
Figure 4, please change the color of simulated measurements of the last iteration to red color for clarity.

Thanks for this suggestion, we have made the change to increase clarity.

Figure 9, please show the snow fall rate from "background" simulation.

The Ku-only retrieval (blue line) can be considered the "background" simulation, since it only uses the Ku-band $Z_{HH}$ measurements and the *a priori* microphysics assumptions to retrieve the PSDs along the ray. This can be thought of as a single Z-S relationship (with attenuation correction applied).